# Tandem hnRNP A1 RNA recognition motifs act in concert to repress the splicing of survival motor neuron exon 7

Irene Beusch[1†], Pierre Barraud[1,2,3*†], Ahmed Moursy[1], Antoine Cléry[1], Frédéric Hai-Trieu Allain[1*]

[1]Institute of Molecular Biology and Biophysics, Eidgenössische Technische Hochschule Zürich, Zürich, Switzerland; [2]Laboratoire de cristallographie et RMN biologiques, UMR 8015, CNRS, Université Paris Descartes, Sorbonne Paris Cité, Paris, France; [3]Laboratoire d'expression génétique microbienne, UMR 8261, CNRS, Université Paris Diderot, Sorbonne Paris Cité, Institut de biologie physico-chimique, Paris, France

**Abstract** HnRNP A1 regulates many alternative splicing events by the recognition of splicing silencer elements. Here, we provide the solution structures of its two RNA recognition motifs (RRMs) in complex with short RNA. In addition, we show by NMR that both RRMs of hnRNP A1 can bind simultaneously to a single bipartite motif of the human intronic splicing silencer ISS-N1, which controls survival of motor neuron exon 7 splicing. RRM2 binds to the upstream motif and RRM1 to the downstream motif. Combining the insights from the structure with in cell splicing assays we show that the architecture and organization of the two RRMs is essential to hnRNP A1 function. The disruption of the inter-RRM interaction or the loss of RNA binding capacity of either RRM impairs splicing repression by hnRNP A1. Furthermore, both binding sites within the ISS-N1 are important for splicing repression and their contributions are cumulative rather than synergistic.

**\*For correspondence:** pierre.barraud@cnrs.fr (PB); allain@mol.biol.ethz.ch (FH-TA)

[†]These authors contributed equally to this work

**Competing interests:** The authors declare that no competing interests exist.

## Introduction

Post-transcriptional control of gene expression is a tightly regulated process in higher organisms, which involves a large variety of RNA-binding proteins that associate with newly transcribed messenger RNAs. Among them, hnRNP A1, one of the most abundant and ubiquitously expressed nuclear proteins, participates in a wide range of nucleic acid processing events, and hence constitutes a major regulator of gene expression (reviewed in [*He and Smith, 2009*; *Jean-Philippe et al., 2013*]). For instance, hnRNP A1 participates in the processing of micro-RNA precursors (*Guil and Cáceres, 2007*; *Michlewski et al., 2008*; *Choudhury and Michlewski, 2012*), in the biogenesis and maintenance of telomeres (*LaBranche et al., 1998*; *Zhang et al., 2006*; *Flynn et al., 2011*), in transcription regulation (*Fukuda et al., 2002*; *Campillos et al., 2003*) and in nucleo-cytoplasmic mRNA transport (*Piñol-Roma and Dreyfuss, 1992*; *Izaurralde et al., 1997*; *Reed and Hurt, 2002*). However, the best-characterized function of hnRNP A1 resides in its extensive implication in constitutive and alternative splicing, where it generally acts as a splicing repressor in enhancing exon skipping (*Mayeda and Krainer, 1992*; *Cáceres et al., 1994*; *Yang et al., 1994*).

HnRNP A1 plays a decisive role in the splicing regulation of exon 7 of the survival motor neuron (*SMN*) genes. This splicing event is critically associated with the neuromuscular disease spinal muscular atrophy (SMA) and is one of the most studied examples of a splicing regulated exon. SMA originates from a deletion or a mutation within the *SMN1* gene, which impairs the production of the SMN protein. The *SMN2* gene is nearly identical to *SMN1* but fails to rescue the inactive *SMN1*

because a critical C-to-U change at position 6 of exon 7 strongly weakens exon 7 splicing (*Lefebvre et al., 1995*; *Cartegni and Krainer, 2002*). This C-to-U change transforms an exonic splicing enhancer (ESE) into an exonic splicing silencer (ESS) (*Cartegni and Krainer, 2002*; *Kashima and Manley, 2003*). As a consequence, the large majority (80%) of the transcripts originating from *SMN2* lacks exon 7, resulting in an unstable protein product. The gene still produces 20% full-length transcripts and hence some functional SMN protein necessary for survival (*Lefebvre et al., 1995*, *1997*). The *SMN2* gene is a major modulator of the severity of SMA. Increasing the amount of functional SMN protein by restoring an *SMN1*-like splicing pattern in *SMN2* is a very promising approach for SMA treatments (*Foust et al., 2010*; *Hua et al., 2010*). Anti-sense oligonucleotides (ASO) targeting a strong intronic splicing silencer (ISS) at the beginning of intron 7 (named ISS-N1), which contains a bipartite motif bound by hnRNP A1, have been shown to enhance *SMN2* exon 7 inclusion in cell culture (*Hua et al., 2007*) and in SMA mouse models (*Hua et al., 2011*, *2015*). This ASO has successfully passed a clinical research program for the treatment of children with SMA (*Chiriboga et al., 2016*) and has recently been approved by the FDA for treatment of SMA licensed under the name SPINRAZA (Nusinersen).

The mechanisms by which hnRNP A1 participates in splicing are numerous (*Mayeda and Krainer, 1992*; *Blanchette and Chabot, 1999*; *Eperon et al., 2000*; *Okunola and Krainer, 2009*; *Tavanez et al., 2012*). It involves direct binding along the mRNA precursor (pre-mRNA) by specific recognition of *cis*-acting elements. This is achieved by the N-terminal part of the protein, also referred to as unwinding protein 1 (UP1), which contains two closely related RNA recognition motifs (RRM1 and RRM2) responsible for the specific RNA-binding properties (*Burd and Dreyfuss, 1994*; *Maris et al., 2005*). The RRMs are followed by a glycine-rich C-terminal region (G-domain) with both protein- and RNA-binding properties (*Kumar et al., 1990*; *Cartegni et al., 1996*). While not essential for RNA binding, the G-domain has been shown to be critical for correct splice site selection (*Blanchette and Chabot, 1999*) and to be involved in liquid droplet formation (*Lin et al., 2015*; *Molliex et al., 2015*). Moreover, it is required for cooperative RNA binding (*Zhu et al., 2001*) that can be directional in the 3′−5′direction and extend over tens of bases (*Okunola and Krainer, 2009*).

Although the two RRMs are highly alike (35% identical and 60% similar), they are neither redundant nor functionally equivalent. Indeed, chimeric proteins constructed by duplication, swapping or deletion of the RRMs affect hnRNP A1 alternative splicing properties differently (*Mayeda et al., 1998*). However, it is unclear which specific properties render hnRNP A1 RRMs non-equivalent. So far, hnRNP A1 RRMs have not been studied independently, which impeded a proper description of their individual particularities essential for understanding hnRNP A1 behavior in splicing.

HnRNP A1 RRMs are interacting with one another and adopt a single defined orientation, as observed by X-ray crystallography and solution NMR (*Shamoo et al., 1997*; *Xu et al., 1997*; *Barraud and Allain, 2013*). Since RRMs are asymmetric binding platforms (*Maris et al., 2005*), the mode of RNA binding is strongly restricted by the relative orientation of the RRMs (reviewed in [*Barraud and Allain, 2013*]). Structural insights into the RNA-binding mode of hnRNP A1 are mostly derived from the structure of UP1 bound to single-stranded telomeric DNA repeats (*Ding et al., 1999*). In this structure, two symmetry-related molecules of UP1 interact to form a dimer that binds two strands of DNA in an anti-parallel manner, each strand extending across the dimer interface (*Appendix 1—figure 1*). This peculiar topology could be the result of crystal packing forces or might only be relevant for the binding of hnRNP A1 to telomeric DNA repeats. While a recent structure of UP1 in complex with an RNA trinucleotide has given insights into RNA recognition by hnRNP A1 RRM1, no RNA was observed bound to RRM2 (*Morgan et al., 2015*). Hence the question of the RNA-binding topology of hnRNP A1 RRMs in solution remains open, especially when encountering a long RNA. In that case, it remains to be determined whether the two RRMs can bind simultaneously to the same pre-mRNA without dimerization.

To address those questions, we studied each RRM of hnRNP A1 in isolation and showed that both RRMs have similar sequence-specificity and affinity. But while RRM2 specifically recognizes three nucleotides, RRM1 is able to recognize a longer motif. We also investigated how hnRNP A1 RRMs bind the ISS-N1 RNA and showed that both RRMs can bind this RNA simultaneously, RRM2 binding its 5′part. This RNA-binding topology directly depends on the relative orientation of the two RRMs and we showed that disrupting the inter-RRM interface impairs hnRNP A1 function in splicing. We also showed that both RRM binding sites within ISS-N1 are important for *SMN* splicing repression and that their binding contributions are cumulative rather than synergistic.

## Results

### Initial binding studies of hnRNP A1 RRMs with small RNA motifs

In order to better characterize the differences in terms of specificities of each RRM, we decided to study each hnRNP A1 RRM in isolation. We thus performed NMR titrations of the isolated RRMs with several short RNA ranging from 6 to 8 nucleotides, each containing a core AG dinucleotide central for hnRNP A1 binding (*Abdul-Manan and Williams, 1996*) (*Appendix 1—table 1*). The RNA sequences were derived from the SELEX motifs for full-length hnRNP A1 or for isolated RRMs (*Burd and Dreyfuss, 1994*) as well as from known hnRNP A1 binding sites (*Hua et al., 2008*). Interestingly, among the different RNA sequences that we tested, we could find for each RRM one RNA sequence where complex formation is in the slow exchange regime, indicative of a strong binding (*Dominguez et al., 2011*). For RRM1 this was observed with 5′-UUAGGUC-3′ (*Figure 1*, and *Figure 1—figure supplement 1*), and for RRM2 with 5′-UCAGUU-3′ (*Figure 2*, and *Figure 2—figure supplement 1*). These RNA sequences differ slightly from the SELEX high affinity selected sequence (5′-UAGGGA/U-3′) (*Burd and Dreyfuss, 1994*) obtained with full-length hnRNP A1, and from the single-stranded DNA motif of telomeric repeats (*Ding et al., 1999*; *Myers and Shamoo, 2004*). Note that our binding sequence for RRM2 is found at the 3′ end of several of the selected sequences obtained with a SELEX done with RRM2 in isolation (*Burd and Dreyfuss, 1994*). We therefore decided to structurally characterize hnRNP A1 RRMs in complex with these two RNAs binding in slow exchange.

### NMR structure of hnRNP A1 RRM1 in complex with 5′-UUAGGUC-3′

We solved the solution structure of hnRNP A1 RRM1 in complex with 5′-UUAGGUC-3′ using 2560 NOE-derived distance restraints including 117 intermolecular ones. This large number of constraints allowed us to obtain a precise structure with a backbone r.m.s.d. over the entire domain of $0.40 \pm 0.11$ Å for the 20 conformers ensemble (*Figure 1A* and *Table 1*).

The $\beta$-sheet surface of RRM1 and the residues of the inter-domain linker are accommodating five nucleotides ($U_1$ to $G_5$), four of which are sequence-specifically recognized through a set of intermolecular hydrogen bonds to the Watson-Crick edges of the bases (*Figure 1B–F*). Overall, the structure is similar to the crystal structures of UP1 bound to (5′-TTAGGG-3′)$_2$ (*Ding et al., 1999*; *Myers and Shamoo, 2004*). The central $A_3G_4$ dinucleotide stacks onto Phe17 and Phe59 of the RNP2 and RNP1 motifs, and is specifically recognized via hydrogen bonds with the backbone atoms of Arg88 and Val90 (*Figure 1D,E*). $A_3$ adopts an *anti* whereas $G_4$ adopts a *syn* conformation similarly to the conformations observed in the UP1-DNA complexes. However, this differs from the recently reported crystal structure of UP1 RRM1 bound to a short RNA (5′-AGU-3′) (*Morgan et al., 2015*), where the adenine adopts a *syn* conformation which could be influenced by the fact that the adenine is at the 5′-end of the oligonucleotide. Importantly, the particular orientation of the $A_3G_4$ dinucleotide onto RRM1 positions the 2-amino group of $G_4$ to hydrogen-bond with the 2′-hydroxyl oxygen of $A_3$ (*Figure 1—figure supplement 2A*). This intramolecular hydrogen bond might be in part responsible for the pronounced preference of both hnRNP A1 and UP1 for single-stranded RNA versus DNA (*Nadler et al., 1991*). In 10 conformers of the ensemble, the side chain of Arg55 is contacting the phosphate group bridging $A_3$ and $G_4$, similarly to the UP1-DNA structure (*Ding et al., 1999*). The flanking nucleotides $U_2$ and $G_5$ are recognized via interactions with the side chains of Glu85 and Lys87 for $U_2$, and Asp42 and Arg92 for $G_5$ (*Figure 1C,F*). Although $U_1$ is not specifically recognized, its position is relatively well defined in the structural ensemble (*Figure 1A*), possibly due to an interaction with the aromatic ring of Phe23 (*Figure 1B*). The nucleotides at the 3′ end, $U_6$ and $C_7$, are not interacting with RRM1 and are therefore highly disordered in the structural ensemble.

### NMR structure of hnRNP A1 RRM2 in complex with 5′-UCAGUU-3′

We solved the solution structure of hnRNP A1 RRM2 in complex with 5′-UCAGUU-3′ with 2253 NOE-derived distance restraints including 111 intermolecular ones. As for RRM1, we obtained a precise structure with a backbone r.m.s.d. of $0.45 \pm 0.08$ Å for the 20 conformers ensemble (*Figure 2A* and *Table 1*).

The $\beta$-sheet surface of RRM2 with the residues of the C-terminal end of the domain are accommodating three nucleotides ($C_2$ to $G_4$), two of which are sequence-specifically recognized through a

**Table 1.** NMR experimental restraints and structural statistics.

| | RRM1+UUAGGUC | | RRM2+UCAGUU | |
|---|---|---|---|---|
| | **Protein** | **RNA** | **Protein** | **RNA** |
| *Distance restraints* | | | | |
| Total NOE (intramolecular) | 2356 | 87 | 2083 | 59 |
| Intra-residue | 395 | 63 | 369 | 45 |
| Sequential | 609 | 24 | 450 | 14 |
| Medium range (\|i-j\|<5 residues) | 522 | 0 | 424 | 0 |
| Long range (\|i-j\|≥5 residues) | 830 | 0 | 840 | 0 |
| Hydrogen bonds (intramolecular) | 22 | 0 | 21 | 0 |
| Protein-RNA intermolecular NOE | 117 | | 111 | |
| Protein-RNA intermolecular hydrogen bonds | 2 | | 2 | |
| *Distance restraints violations (mean ± s.d.)* | | | | |
| Number of NOE violations > 0.2 Å | 1.9 ± 1.2 | | 3.1 ± 1.0 | |
| Maximum NOE violation (Å) | 0.25 ± 0.05 | | 0.27 ± 0.04 | |
| *Dihedral angle restraints* | | | | |
| Sugar pucker | | 4 | | 4 |
| *Dihedral violations (mean ± s.d.)* | | | | |
| Number of dihedral violations > 5° | | 0 | | 0 |
| Maximum dihedral violation (°) | | 0.04 ± 0.16 | | 0.47 ± 0.86 |
| *R.m.s.d. from mean structure (Å)* | | | | |
| Protein* | | | | |
| Backbone | 0.40 ± 0.11 | | 0.45 ± 0.08 | |
| Heavy atoms | 0.83 ± 0.10 | | 0.78 ± 0.07 | |
| RNA[†] | | | | |
| Heavy atoms | 0.62 ± 0.13 | | 0.78 ± 0.14 | |
| *Deviation from ideal geometry (mean ± s.d.)* | | | | |
| Bond lengths (Å) | 0.0037 ± 0.0001 | | 0.0035 ± 0.0001 | |
| Bond angles (°) | 0.51 ± 0.02 | | 0.49 ± 0.01 | |
| Impropers (°) | 1.13 ± 0.09 | | 1.22 ± 0.06 | |
| *Ramachandran analysis* | | | | |
| Most favored region | 85.9% | | 83.1% | |
| Allowed region | 13.6% | | 16.6% | |
| Disallowed region | 0.5% | | 0.2% | |
| *CING Red/Orange/Green scores* | | | | |
| R/O/G (%) | 18/28/54 | | 21/25/54 | |

*Protein r.m.s.d. was calculated using residues 11–92 for RRM1 and 103–112, 115–139, 144–187 for RRM2;
[†]RNA r.m.s.d. was calculated using residues 2–5 for RNA bound to RRM1 and 2–4 for the RNA bound to RRM2.

set of intermolecular hydrogen-bonds (*Figure 2B–E*). Overall, the structure is similar to UP1 RRM2 bound to (5′-TTAGGG-3′)$_2$ (*Ding et al., 1999*; *Myers and Shamoo, 2004*). The central A$_3$G$_4$ dinucleotide stacks onto Phe108 and Phe150 of the RNP motifs and is recognized via a set of hydrogen-bonds from the backbone atoms of Lys179 and Leu181 (*Figure 2D,E*). A$_3$ adopts an *anti* conformation whereas G$_4$ is *syn*. Similarly to RRM1, the 2-amino group of G$_4$ forms an hydrogen-bond with the 2′-hydroxyl oxygen of A$_3$ (*Figure 1—figure supplement 2B*). In 14 structures of the ensemble, Arg146 is contacting the phosphate group bridging A$_3$ and G$_4$, as seen in the crystal structures of UP1-DNA (*Ding et al., 1999*). In addition, the side chain of Met186 in helix α3 stacks over the A$_3$

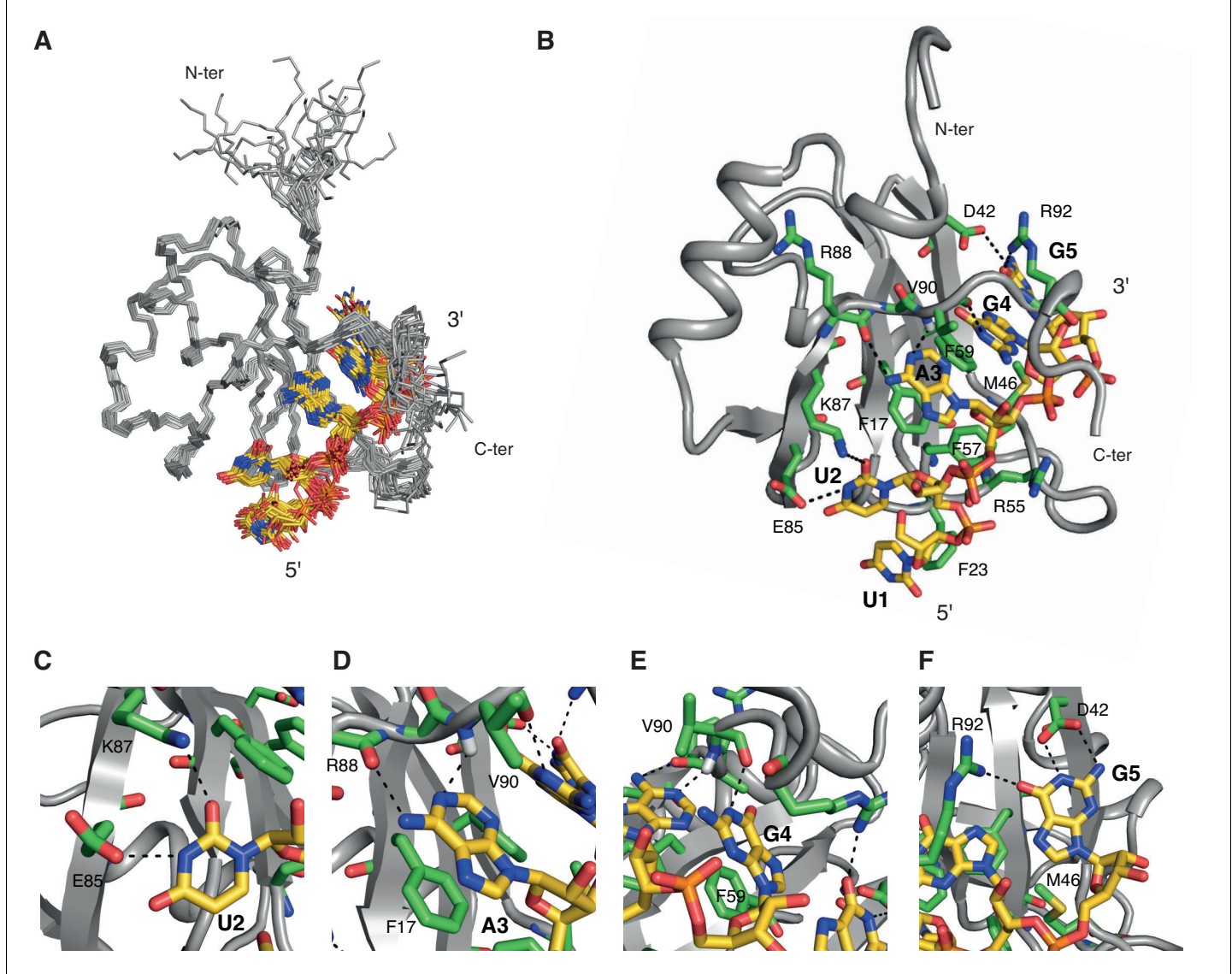

**Figure 1.** Overview of the solution structure of hnRNP A1 RRM1 bound to 5′-UUAGGUC-3′ RNA. (**A**) NMR ensemble. Overlay of the 20 final structures superimposed on the backbone of the structured part (11-92) and represented as a ribbon trace (N, Cα, C′). The not-well-defined nucleotides $U_6$ and $C_7$ are omitted for a better overview. The RNA is shown in stick representation with the carbon atoms in yellow, nitrogen in blue, phosphate in orange and oxygen in red. (**B**) Cartoon drawing of a representative structure of the NMR ensemble. Residues with important roles in RNA binding are shown as sticks, with the carbon atom in green. All other atoms have the same color code as in panel **A**. (**C–F**) Close-up views of each single nucleotide recognition by hnRNP A1 RRM1. Representation and colors are similar to panel **B**. Residues with important roles in RNA binding are labeled.

The following figure supplements are available for figure 1:

**Figure supplement 1.** Overview of the titration of 5′-UUAGGUC-3′to hnRNP A1 RRM1.

**Figure supplement 2.** Intramolecular hydrogen bond within the AG dinucleotide constituting the core of the hnRNP A1 RRMs binding motif.

ring. This base is therefore sandwiched between Phe108 and Met186. Note that helix α3(Lys183-Ser191) folds into an α-helix only upon binding to RNA. This folding event explains the large chemical shift variations seen for this entire region upon RNA binding (*Figure 2—figure supplement 2A*). This event is driven by the Met186, as shown by the much smaller changes found with a M186A variant (*Figure 2—figure supplement 2B*). The upstream nucleotide $C_2$ is not specifically recognized

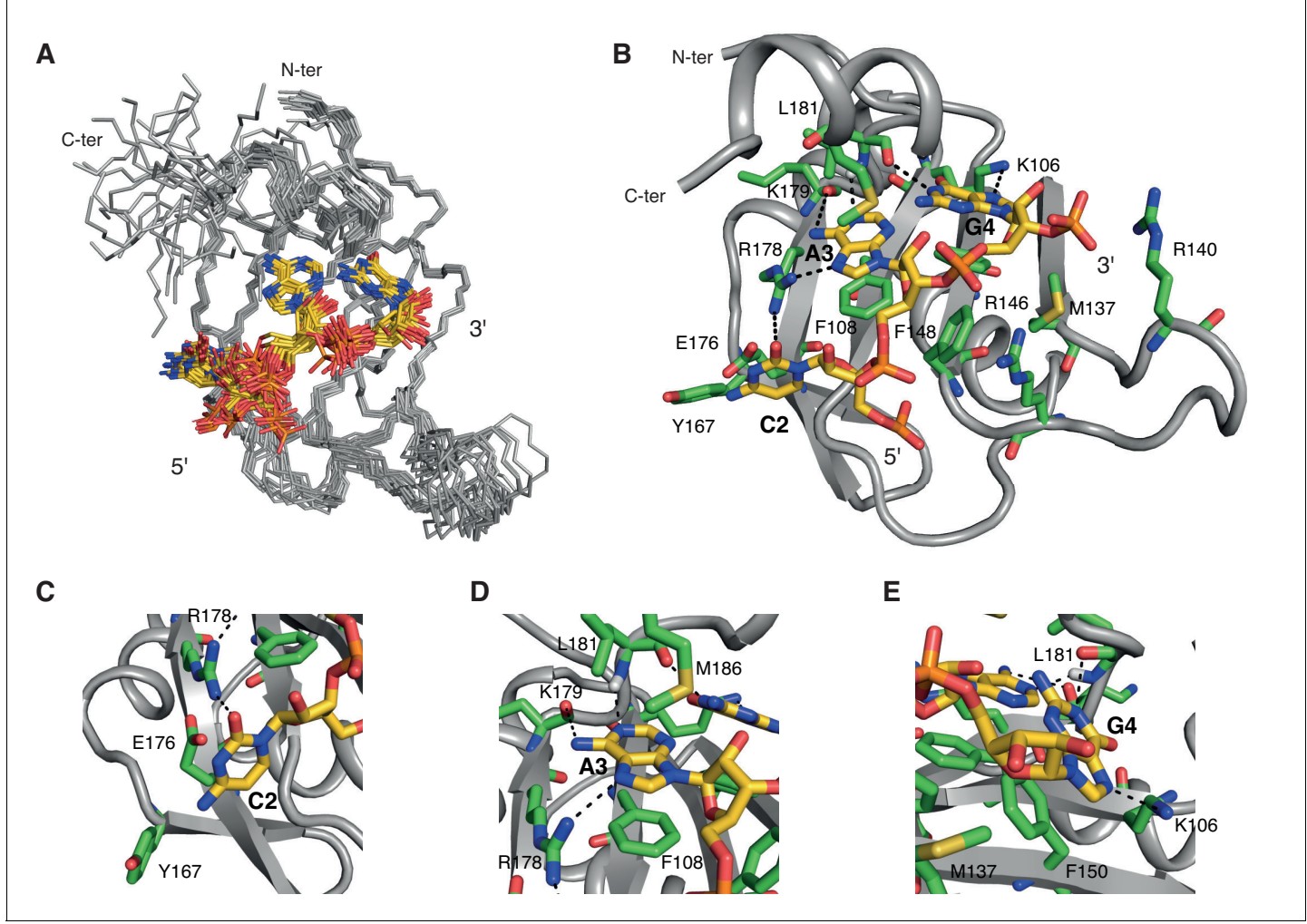

**Figure 2.** Overview of the solution structure of hnRNP A1 RRM2 bound to 5′-UCAGUU-3′ RNA. (**A**) NMR ensemble. Overlay of the 20 final structures superimposed on the backbone of the structured part (103-112, 115-139, 144-187) and represented as a ribbon trace (N, Cα, C′). The not-well-defined N- and C-terminal residues as well as nucleotides $U_1$, $U_5$ and $U_6$ are omitted for a better overview. The RNA is shown in stick representation with the carbon atoms in yellow, nitrogen in blue, phosphate in orange and oxygen in red. (**B**) Cartoon drawing of a representative structure of the NMR ensemble. Residues with important roles in RNA binding are shown as sticks, with the carbon atom in green. All other atoms have the same color code as in panel **A**. (**C–E**) Close-up views of each single nucleotide recognition by hnRNP A1 RRM2. Representation and colors are similar to panel **B**. Residues with important roles in RNA binding are labeled.

The following figure supplements are available for figure 2:

**Figure supplement 1.** Overview of the titration of 5′-UCAGUU-3′ to hnRNP A1 RRM2.

**Figure supplement 2.** The side chain of methionine 186 drives most of the C-terminal helix α3 folding upon RNA-binding.

but is bound by RRM2 (*Figure 2C*). The $C_2$ ring stacks to both the main-chain of Gly111 and the side-chain of Glu176. Glu176 does not hydrogen bond with $C_2$ unlike its equivalent residue in RRM1. The carbonyl O2 of $C_2$ also interacts with the side chain of Arg178 (*Figure 2C*). In a few conformers of the structural ensemble, the hydroxyl oxygen of Tyr167 is within hydrogen-bond distance from the 4-amino group of $C_2$. This provides a pyrimidine-specific contact since the hydroxyl is capable of acting as a hydrogen-bond donor in case of a uracil. Finally, $U_1$ is not well defined in the structure unlike what was found in RRM1. The NMR signals from the corresponding region in RRM2 (β1-α1 loop) are absent from both the free and bound spectra, reflecting a conformational dynamic for this

loop. As a consequence, intermolecular contacts between this region and $U_1$ cannot be observed in solution, although intermolecular contacts were seen to the equivalent of $U_1$ in the structure of UP1-DNA (*Ding et al., 1999*). The nucleotides at the 3′-end, $U_5$ and $U_6$ are not interacting with RRM2 and are therefore highly disordered in the structural ensemble. Finally, the side chain of Arg140 is potentially interacting with the phosphate group bridging $G_4$ and $U_5$ (*Figure 2B*), since this contact is observed in eight conformers of the ensemble.

Combining our NMR study with the known structural and biochemical information on hnRNP A1 RNA-binding (*Burd and Dreyfuss, 1994*; *Abdul-Manan and Williams, 1996*; *Ding et al., 1999*; *Myers and Shamoo, 2004*; *Bruun et al., 2016*), we can decipher that the optimal RNA-binding motifs should be: 5′-YAGG-3′ and 5′-YAGN-3′ for RRM1 and RRM2, respectively, where Y is a pyrimidine and N any nucleotide. Interestingly, motifs of in vivo targets of hnRNP A1 can be fairly different (see for instance the ISS-N1 binding sites 5′-CAGCAU-3′ and 5′-UGAAAG-3′ in the context of *SMN* pre-mRNAs [*Hua et al., 2008*] or the binding sites in *SKA2* exon 3 recently identified by iCLIP [*Bruun et al., 2016*]).

## RNA specificities and preferences of each individual RRM of hnRNP A1

Therefore, in order to further confirm or infirm our NMR-derived RNA-binding consensus for each RRM, we measured the affinity of the isolated RRM1 and RRM2 for RNA sequences varying at the two positions flanking the AG core using ITC. In addition, in order to quantitatively estimate the importance of the protein-RNA contacts seen in the structures (*Figures 1* and *2*), we mutated few residues responsible for the RNA binding (e.g. RRM2: Arg140, Met186) or for sequence specificity (e.g. RRM1: Asp42, Arg92; RRM2: Tyr167, Glu176). Affinity of wild-type and mutant proteins were measured by ITC with the RNAs used in these structural works (*Table 2*), and with RNAs of slightly different sequences (*Table 3*).

In perfect agreement with the structure of the RRM1-RNA complex, mutating Arg55 to alanine lowers the affinity for the RNA drastically by more than 50-fold (*Table 2*). Mutation of Arg92 or Asp42 also resulted in an affinity decrease by 10- and 3-fold, respectively. This confirmed the RRM1 preference for a guanosine 3′ to the AG core (*Figure 1F* and *Table 3*). Our structural analysis

**Table 2.** Evaluation of residues involved in RNA recognition.

**RRM1 + 5′-UUAGGUC-3′**

| RRM1 | $K_d$ (nM) | Affinity loss | N |
|---|---|---|---|
| wild-type | 292 ± 17 | 1 | 0.93 ± 0.05 |
| | 288 ± 16 | | 0.99 ± 0.05 |
| F23A | 225 ± 13 | 0.7 | 1.07 ± 0.05 |
| | 193 ± 11 | | 0.98 ± 0.05 |
| D42A | 787 ± 42 | 3 | 0.93 ± 0.05 |
| R55A | >10'000 | >50 | 1 |
| R92A | 1000–5000 | >5 | 1.08 ± 0.05 |

**RRM2 + 5′-UCAGUU-3′**

| RRM2 | $K_d$ (nM) | Affinity loss | N |
|---|---|---|---|
| wild-type | 541 ± 33 | 1 | 1.04 ± 0.05 |
| R140A | 1000–5000 | >2 | 1.07 ± 0.05 |
| Y167F | 513 ± 30 | ~1 | 0.99 ± 0.05 |
| E176Q | 237 ± 15 | 0.4 | 1.01 ± 0.05 |
| M186A | 1630 ± 45 | 3 | 0.96 ± 0.05 |

Values are reported as means ± standard error (S.E.). The uncertainties on the fitted parameters were estimated from the data spread and from the uncertainty of the protein concentration determination (5%). $K_d$: dissociation constant in nM. Affinity loss: ratio between $K_d$ of the mutant and $K_d$ of the wild-type. N: number of sites.

**Table 3.** Evaluation of RNA specificity and affinity.

|          | RRM1 WT | RRM1 F23A | RRM1 D42A | RRM1 R92A |
|----------|---------|-----------|-----------|-----------|
| UUAGGUC  | 288 ± 16 (0.99 ± 0.05) | 225 ± 13 (1.07 ± 0.05) | 787 ± 42 (0.93 ± 0.05) | 1000–5000 (1.08 ± 0.05) |
| UAGGUC   | 625 ± 35 (0.96 ± 0.05) | 602 ± 34 (0.91 ± 0.05) | | |
| UUUUAGGUC| 159 ± 9 (0.90 ± 0.05) | 130 ± 8 (0.91 ± 0.05) | | |
| UCAGGUC  | 543 ± 33 (0.90 ± 0.05) | | | |
| UUAGUU   | 3000–9000 (0.97 ± 0.05) | | 1000–5000 (1.07 ± 0.05) | >10'000 (1) |
| UCAGUU   | >10'000 (1) | | | |
| UAAGUU   | 3000–9000 (0.97 ± 0.05) | | | |
| UGAGUU   | 3000–9000 (0.99 ± 0.05) | | | |

|          | RRM2 WT | RRM2 R140A | RRM2 E176Q | RRM2 Y167F | RRM2 M186A |
|----------|---------|------------|------------|------------|------------|
| UCAGUU   | 541 ± 33 (1.04 ± 0.05) | 1000–5000 (1.07 ± 0.05) | 237 ± 15 (1.01 ± 0.05) | 513 ± 30 (0.99 ± 0.05) | 1630 ± 45 (0.96 ± 0.05) |
| UUAGUU   | 125 ± 9 (1.04 ± 0.05) | 433 ± 26 (1.10 ± 0.05) | 286 ± 12 (0.99 ± 0.05) | 118 ± 9 (0.95 ± 0.05) | 253 ± 15 (1.00 ± 0.05) |
| UUAGGUC  | 64 ± 8 (1.09 ± 0.05) | 195 ± 12 (1.01 ± 0.05) | 116 ± 39 (1.01 ± 0.05) | | |
| UCAGGUC  | 129 ± 11 (1.01 ± 0.05) | 862 ± 47 (0.92 ± 0.05) | 132 ± 29 (0.93 ± 0.05) | | |
| UAAGUU   | 543 ± 31 (1.04 ± 0.05) | 1000–5000 (0.97 ± 0.05) | 429 ± 27 (0.94 ± 0.05) | | |
| UGAGUU   | 546 ± 31 (0.93 ± 0.05) | 1000–5000 (1) | 324 ± 20 (1.01 ± 0.05) | | |
| UAGGUC   | 98 ± 8 (1.01 ± 0.05) | 685 ± 40 (1.01 ± 0.05) | 227 ± 14 (1.07 ± 0.05) | | |

Dissociation constant ($K_d$) for RRM1 and RRM2 wild-type and mutants, respectively. Values are reported as means ± standard error (S.E.). The uncertainties on the fitted parameters were estimated from the data spread and from the uncertainties of the protein concentration determination (5%). A dissociation constant range instead of a value is given where no sufficient saturation could be reached for the measurement conditions. The dissociation constants are given in nM. The number of sites N is given in parenthesis.

Source data 1. ITC data for evaluation of RNA specificity and affinity of the RRMs of hnRNP A1.

suggested that Phe23 may contact $U_1$ and contribute to the RNA-binding affinity. Removal of the side-chain (*Table 2*) as well as shortening or elongating the RNA at the 5′ end (*Table 3*), showed that Phe23 does not contribute significantly to RNA binding for these sequences, at least in terms of detectable affinity.

Similarly for RRM2, removing contacts to the phosphate backbone by mutating Arg140 led to a two-fold lower affinity underlining its importance for RNA binding (*Figure 2B* and *Table 2*). Also, confirming the contribution of helix α3 for RNA binding, mutation of Met186 resulted in a 3-fold affinity drop (*Figure 2—figure supplement 2* and *Table 2*). A role for Tyr167 in recognizing $C_2$ could not be confirmed by ITC, since removal of the hydroxyl group (Y167F) has no effect on RNA binding (*Table 2*).

Our structure helps in understanding how RRM2 can accommodate a C at the position preceding the AG core. While a U is specifically recognized by Glu176 via a direct hydrogen-bond to the base (*Ding et al., 1999*; *Myers and Shamoo, 2004*), when a C is present Glu176 stacks under the base (*Figure 2B,C*). Note that as for RRM1, RRM2 prefers a U at this position (see *Table 3*), and our complex with RRM2 was thus not determined with the RNA having the best affinity. It had however the optimal exchange properties in terms of NMR and structure determination (see *Appendix 1—table 1*).

Altogether, our ITC measurements with the isolated RRMs (*Table 2* and *Table 3*) allowed us to propose simple rules regarding the RNA-binding preferences of the RRMs of hnRNP A1. First, both RRMs have a preference for the 5′-UAGG-3′ motif, which was indeed present in many RNAs found by SELEX or iCLIP (*Burd and Dreyfuss, 1994*; *Bruun et al., 2016*). Second, both RRMs are tolerant to accommodate a cytosine 5′ to the AG core, extending the recognition motif to 5′-U/CAGG-3′. Finally, RRM2 is less affected by the replacement of the guanosine 3′ of the AG core than RRM1.

Overall, the optimal recognition motif for RRM1 can be finalized as 5′-U/CAGG-3′ and for RRM2 as 5′-U/CAGN-3′.

Importantly, for all these ITC measurements, RRM2 appears to consistently have a higher affinity than RRM1. However, the available structures of UP1 (*Ding et al., 1999*; *Myers and Shamoo, 2004*; *Morgan et al., 2015*) suggest that the inter-RRM linker (IRL), which is missing from our RRM1 construct, may be equivalent to RRM2 helix α3. We therefore measured the affinity of each RRM in the context of UP1. To do so we prepared mutants of the RNP residues, which are critical for binding the core AG dinucleotide. Mutants UP1-R1r2 (F108A/F150A with RRM2 mutated) and UP1-r1R2 (F17A/F59A with RRM1 mutated) were then titrated with different RNA sequences. In this context, both RRMs have now very similar affinities for all the tested sequences. The affinity of RRM1 for all oligonucleotides was significantly increased (over 5-fold) while RRM2 kept the same affinity for RNA in both contexts (*Table 4*).

Importantly, the mutations were efficient in removing the RNA binding ability of the mutated RRM. After fitting with a 1:1 binding model the obtained N-values of our ITC measurements were robustly ~1. This is indicative of a 1:1 stoichiometry of the complex. Intriguingly, RRM1 in the context of UP1 appears less affected by the replacement of the G following the AG core (~2 fold drop in affinity; *Table 4*) than in the isolated RRM1 construct (at least 10-fold drop in affinity; *Table 4*). We cannot rationalize this point, especially because all structures of hnRNP A1, including ours, show multiple specific contacts of RRM1 with this guanine. Apart from this particular point, the preferences observed for the RRMs in isolation as described above are preserved in the UP1 context.

## Investigating the topology of RNA binding onto hnRNP A1 RRMs

Using ITC and NMR titrations, we have shown that each individual RRM of hnRNP A1 interacts with RNA motifs of 3 to 4 nucleotides and that both RRM1 and RRM2 have a strong preference and a similar affinity for 5′-UAG-3′ in the context of UP1. However, it remains to be determined how the two tandem RRMs of hnRNP A1 organize themselves when binding longer pre-mRNA stretches. Is only a single RRM binding to the RNA or can both RRMs bind simultaneously? Is there a given directionality? We therefore investigated the topology of binding of hnRNP A1 RRMs to a natural intronic splicing silencer containing two adjacent hnRNP A1 binding motifs. We chose the major regulator of *SMN2* pre-mRNA splicing, that is, the so-called ISS-N1, which is present in intron 7 of *SMN2* (*Singh et al., 2006*; *Hua et al., 2008*). It embeds two AG dinucleotides separated by a short spacer of 9 nucleotides (*Figure 3—figure supplement 1*).

We titrated ISS-N1 RNA with UP1 and monitored complex formation with NMR. Although both RRMs are seen to interact with ISS-N1 (*Figure 3—figure supplement 2A*), some signals at the

**Table 4.** Isolated RRMs compared with RRMs in the context of UP1.

| | UP1 R1r2 | | RRM1 | | UP1 r1R2 | | RRM2 | |
|---|---|---|---|---|---|---|---|---|
| | $K_d$ (nM) | N | $K_d$ (nM) | N | $K_d$ (nM) | N | $K_d$ (nM) | N |
| UUAGGUC | 47 ± 2 | 1.02 ± 0.05 | 288 ± 16 | 0.99 ± 0.05 | 56 ± 4 | 1.02 ± 0.05 | 64 ± 8 | 1.09 ± 0.05 |
| UCAGGUC | 171 ± 12 | 0.97 ± 0.05 | 543 ± 33 | 0.90 ± 0.05 | 178 ± 11 | 0.90 ± 0.05 | 129 ± 11 | 1.01 ± 0.05 |
| UUAGUU | 105 ± 8 | 1.05 ± 0.05 | 3000–9000 | 0.97 ± 0.05 | 142 ± 9 | 0.99 ± 0.05 | 125 ± 9 | 1.04 ± 0.05 |
| UCAGUU | 578 ± 40 | 1.11 ± 0.05 | >10'000 | 1 | 637 ± 39 | 1.05 ± 0.05 | 541 ± 33 | 1.04 ± 0.05 |
| UAAGUU | 658 ± 44 | 1.05 ± 0.05 | 3000–9000 | 0.97 ± 0.05 | 685 ± 42 | 1.01 ± 0.05 | 543 ± 31 | 1.04 ± 0.05 |
| UGAGUU | 654 ± 50 | 1.09 ± 0.05 | 3000–9000 | 0.99 ± 0.05 | 625 ± 41 | 1.04 ± 0.05 | 546 ± 31 | 0.93 ± 0.05 |

Values are reported as means ± standard error (S.E.). The uncertainties on the fitted parameters were estimated from the data spread and from the uncertainty of the protein concentration determination (5%). A $K_d$ range instead of a value is given where no sufficient saturation could be reached for the measurement conditions. $K_d$: dissociation constant in nM. N: number of sites.

**Source data 1.** ITC measurements of protein variants with 5′-UUAGUU-3′.

**Source data 2.** ITC data for evaluation of RNA specificity and affinity of the RRMs of hnRNP A1 in the context of UP1.

binding interface disappeared upon RNA addition. This is indicative of a complex in intermediate exchange for which structure determination is difficult to achieve (*Dominguez et al., 2011*). We then optimized the sample towards a more favorable exchange regime for structure determination by NMR by minimally mutating either the RNA, the protein, or both, taking advantage of our work with the individual RRMs (*Table 4*).

Briefly, we introduced guanines downstream of the AG core dinucleotide to increase the affinity (*Table 4*), creating the RNA mutants ISS-N1-c14g, ISS-N1-u25g, and ISS-N1-14g25g (*Figure 3—figure supplement 1*). Alone, this approach was not sufficient to recover narrow NMR line-widths (See *Figure 3—figure supplement 2* and Appendix 2 for details). Since it is not necessarily the complex with the best affinity that will give the best NMR signals, we then took the opposite approach of weakening the overall affinity of the complex with the protein mutant UP1-R140A (*Table 3*). Combining this protein mutant with the previous RNA mutants gave NMR spectra of much improved quality (*Figure 3A–D*), especially for the double mutant ISS-N1-14g25g and the single mutant ISS-N1-u25g. In addition, a detailed analysis of these NMR titrations (*Figure 3E–H*) strongly suggested that RRM1 binds to the 3′-motif of ISS-N1 irrespective of the point mutations introduced in the RNA and therefore RRM2 binds the 5′-motif (See Appendix 2 for details).

## Structural investigation of ISS-N1 bound to hnRNP A1 RRMs

Next, we structurally investigated the protein-RNA complex formed between UP1-R140A and ISS-N1-u25g by NMR. Importantly, the pattern of chemical shift changes observed for UP1 binding to one equivalent of ISS-N1 (*Figure 4—figure supplement 1B*) is highly similar to those observed for the individual RRMs bound to short RNA oligonucleotides (*Figure 1—figure supplement 1B* and *Figure 2—figure supplement 1B*). This shows that only one molecule of hnRNP A1 binds and confirms the involvement of both RRMs of hnRNP A1 in ISS-N1 binding. The binding mode resembles the one determined for RRMs in isolation (*Figures 1* and *2*).

We could reach almost complete resonance assignment on the protein (See Appendix 2), and could confidently assign the RNA around the two AG dinucleotides present in the ISS-N1 sequence (*Appendix 2—table 1*). Overall we observed 87 unambiguous intermolecular protein-RNA contacts with the assigned nucleotides (*Appendix 2—table 2*, *Figure 4—figure supplements 2* and *3*). Unfortunately, due to severe resonance overlap we were limited in our assignments of the RNA resonances. Nevertheless, the obtained network of intermolecular contacts clearly supports the topology of RNA binding suggested by the NMR titrations. Indeed, the residues from RRM1 display many intermolecular contacts to the 3′-end of ISS-N1, namely to $A_{22}A_{23}G_{24}G_{25}$ and residues from RRM2 to the 5′-end, namely $C_{11}A_{12}G_{13}$ (*Appendix 2—table 2*).

However, the determined RNA-binding topology depends strongly on the correctness of the RNA assignments, which were only partial. Hence, to independently confirm our results, we measured paramagnetic relaxation enhancement (PRE) experiments with a nitroxide spin-label attached to the 3′-end of the RNA. For this purpose, we left the two RNA binding motifs unchanged, namely the $C_{11}A_{12}G_{13}$ motif for RRM2 and the $A_{22}A_{23}G_{24}G_{25}$ motif for RRM1, and placed a 4-thio-uridine spin-label downstream of $G_{25}$. The spin-label should not perturb RNA binding at this position, but it should affect the resonances of the RRM binding the 3′-AG motif. We know from our structures of the single RRMs, that the spin-label should be capable of sampling multiple conformations resulting in the perturbation of a large area on one side of the domain. In addition, to exclude that the R140A mutation influences the binding topology, the PRE data were also recorded for the modified ISS-N1 bound by UP1 wild-type (*Figure 4A,B*).

We could confidently transpose the backbone assignment from the UP1-R140A:ISS-N1-u25g complex to most resonances of the present complexes and therefore have unambiguous probes all over the UP1 structure. Comparison of the spectra of the oxidized *vs.* reduced spin label shows that the signals affected by the presence of the paramagnetic probe originated mostly from RRM1 and the interdomain linker matching perfectly our previous experimental data. We observe this local perturbation in RRM1 and the interdomain linker for both UP1 constructs (*Figure 4A–D*). Altogether, this constitutes an independent evidence that UP1 binds to the ISS-N1 using both RRMs, and that RRM1 binds to the 3′ AG-motif and RRM2 to the 5′ AG-motif, as schematically represented on *Figure 4E*.

To our knowledge, these results represent the first clear demonstration that two RRMs of the same hnRNP A1 molecule can bind the same RNA molecule simultaneously. Combining all the

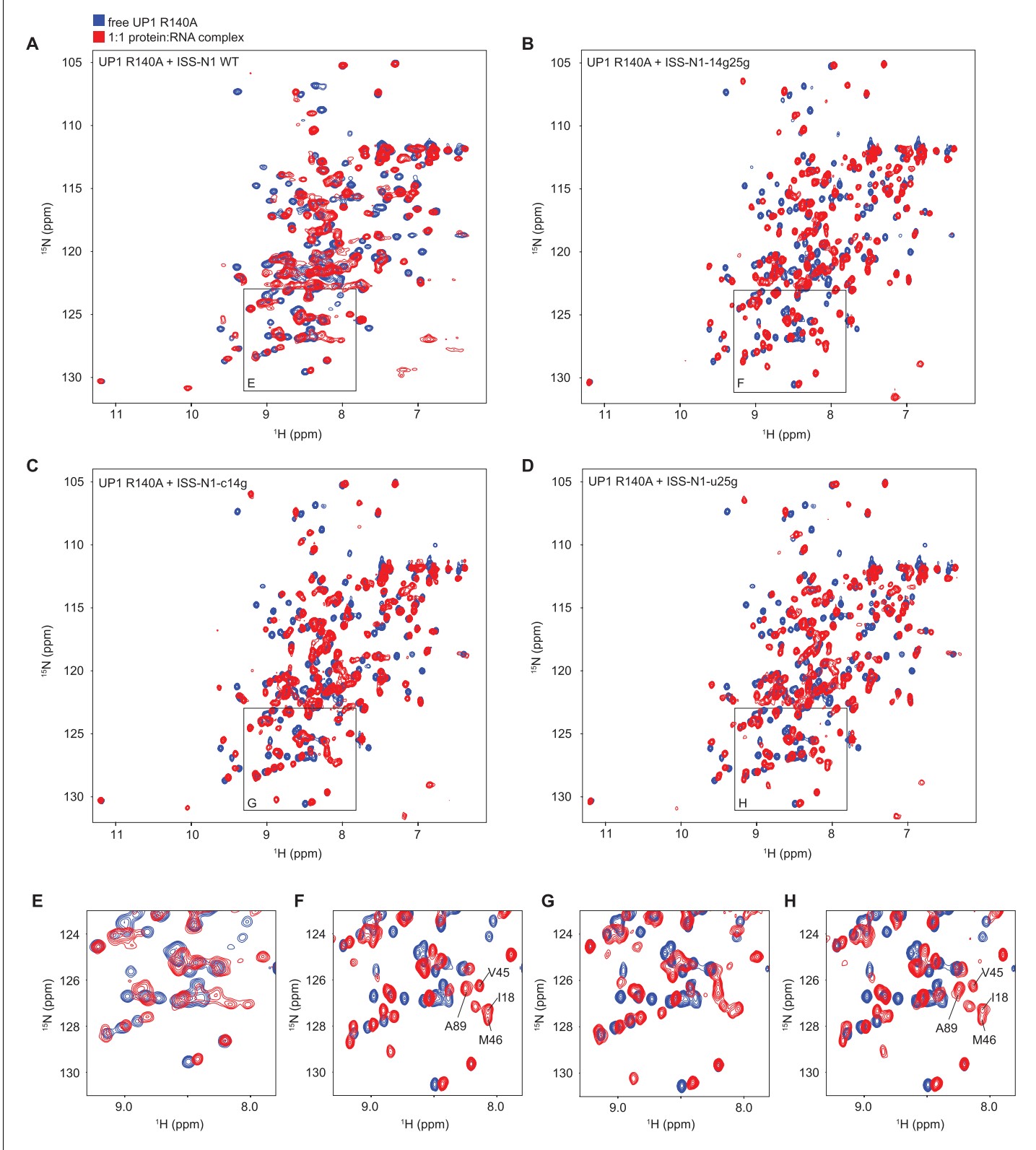

**Figure 3.** Titration of ISS-N1 variants to UP1 R140A variant. Overlays of 2D ($^{15}$N,$^{1}$H)-HSQC spectra of the free UP1 R140A variant (blue) and UP1 R140A in the presence of 1 equivalent of ISS-N1 variants (red). The region shown in a close-up is shown on the respective panel (**E–H**). Residues pointed out for RRM1 in the text are marked in panels (**F**) and (**H**). NMR signals can be obtained at a good linewidth for the complex of UP1 R140A + ISS-N1-u25g.

*Figure 3 continued on next page*

Figure 3 continued

(A) UP1 R140A (0.3 mM) + ISS-N1 WT. (B) UP1 R140A (0.3 mM) + ISS-N1-14g25g. (C) UP1 R140A (0.3 mM) + ISS-N1-c14g. (D) UP1 R140A (0.3 mM) + ISS-N1-u25g. (E–H) Close-up view of the region boxed on panels A–D).

The following figure supplements are available for figure 3:

**Figure supplement 1.** Sequence and localization of the *cis*-acting element ISS-N1 in the *SMN2* intron 7.

**Figure supplement 2.** Titration of ISS-N1 variants to UP1 wild-type.

structural information obtained on this complex, we could build a structural model showing how hnRNP A1 RRMs can assemble on the same pre-mRNA stretch (see Appendix 2 for details). Our structural model (*Figure 4F*) differs strongly from the crystal structure of hnRNP A1 RRMs bound to telomeric DNA repeats, where hnRNP A1 dimerizes and binds to two distinct DNA oligonucleotides (*Appendix 1—figure 1*), and where RRM1 binds the 5′-end of a DNA molecule and RRM2 the 3′-end of another DNA strand (*Ding et al., 1999*). With our structural model, the antiparallel organization of hnRNP A1 RRM1 and RRM2 imposes a looping of the RNA molecule, such as suggested for other hnRNP family members with interacting RRMs adopting a single defined orientation (reviewed in [*Barraud and Allain, 2013*]). To adopt this mode of binding, the RNA spacer between the 3′- and 5′-motifs must be sufficiently long. In the case of the ISS-N1, the RNA spacer between the two AG motifs is 9-nucleotide long (*Figure 3—figure supplement 1*). With various calculations of structural models where we reduced the length of the RNA spacer, we could see that due to topological constraints, the RNA spacer between the two AGs must be at least 4-nucleotide long (see Appendix 2 for details).

## The relative orientation of the hnRNP A1 RRMs is essential for its function in cells

Previous research has underlined the importance and regulatory function of the ISS-N1 for *SMN2* splicing (*Singh et al., 2006*; *Hua et al., 2008*), and while it has been shown that the two RRMs of hnRNP A1 have distinct roles in alternative splicing (*Mayeda et al., 1998*), our structural work does not prove that the two RRMs effectively bind the bipartite ISS-N1 in vivo.

Our structural investigation of the mode of binding of the ISS-N1 by hnRNP A1 hints at the importance of both RRMs for interacting with the RNA. For RRM1 to bind with high affinity, the IRL must be positioned such as to sandwich the RNA together with the $\beta$-sheet surface of RRM1. This proper positioning of the IRL is achieved by RRM2 contacting RRM1 through a network of interactions including two salt bridges (Arg88-Asp157 and Arg75-Asp155)(*Ding et al., 1999*; *Barraud and Allain, 2013*; *Morgan et al., 2015*). If the inter-RRM orientation is linked to the function, any disruption of this interaction should impair hnRNP A1 splicing regulation. We therefore prepared an R75A/D157K double-mutant (hnRNP A1 INT1), as well as a D155R/D157K double-mutant (INT2) of FLAG-tagged hnRNP A1. This latter mutant was extended to include an I164A mutation (INT3), which alters the hydrophobic contacts on one end of the interface. For the INT1 double-mutant, we chose not to mutate Arg75 to Asp or Glu, as this might not preserve the cis-Pro76 (*Pal and Chakrabarti, 1999*). Using NMR, we could show that the mutations interfere with the RRMs' interactions. The determined correlation time $\tau_c$ suggests a significant disruption of the interface (*Figure 5—figure supplement 1*), which is also reflected by the chemical shift changes observed at the RRM interfaces for the mutants (*Figure 5—figure supplement 1*). Nevertheless, the mutations do not render the RRMs fully independent (the changes in $\tau_c$ are modest), possibly due to a strong topological constraint imposed by the IRL.

Wild-type and mutant hnRNP A1 constructs were co-transfected together with an *SMN1* mini-gene (*Hua et al., 2007*) into HEK293T cells. After the RNA was extracted, *SMN1* exon 7 inclusion percentage was quantified by RT-PCR. We chose the *SMN1* minigene system (*Lefebvre et al., 1995*; *Frugier et al., 2002*), where exon 7 inclusion occurs by default at 93% (*Figure 5A*, lane 1) and overexpression of hnRNP A1 can lower exon 7 inclusion down to 6% (*Figure 5A*, lane 2). This large response upon hnRNP A1 overexpression allows to catch impairment in hnRNP A1 function. Nuclear

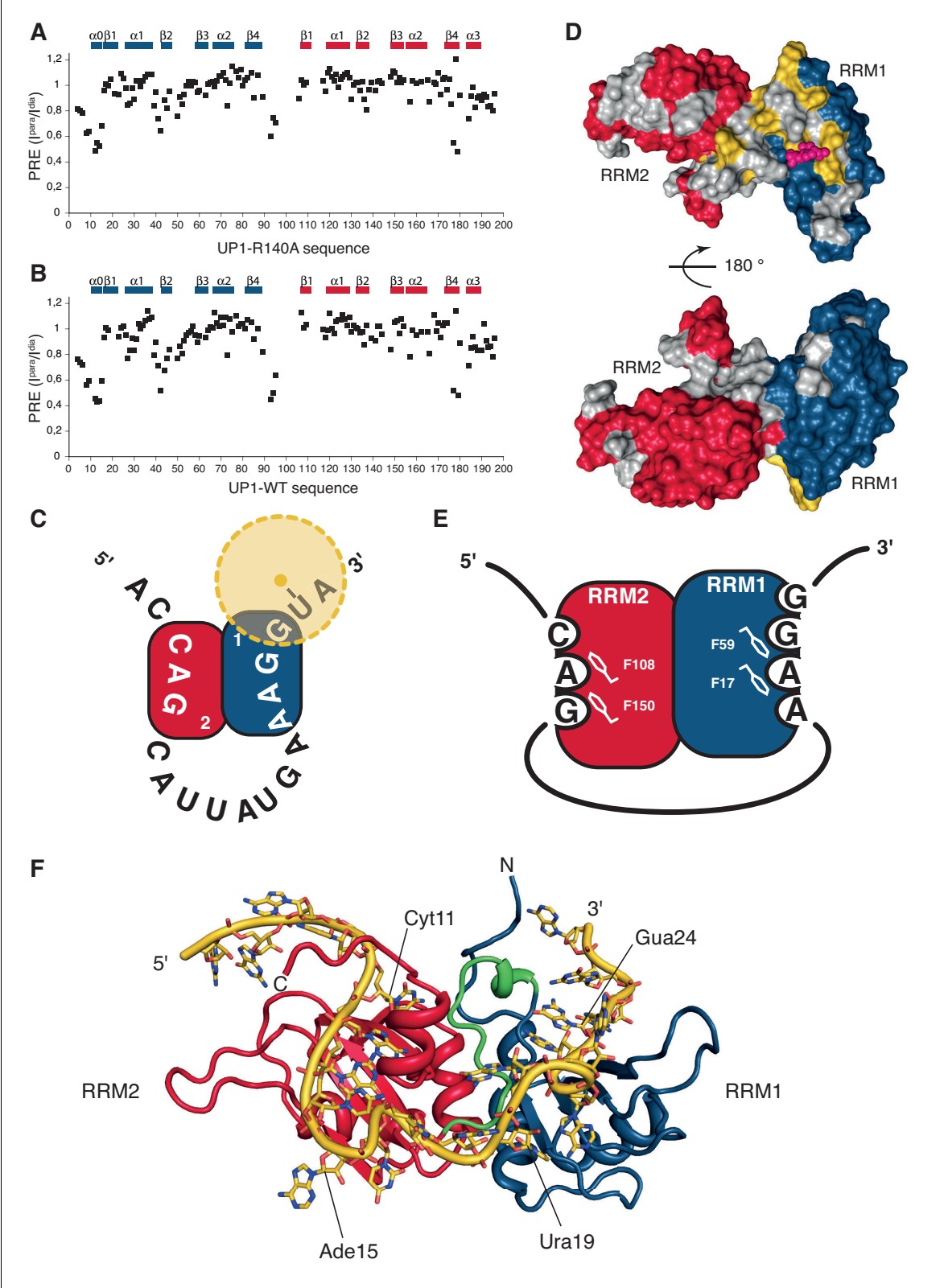

**Figure 4.** HnRNP A1 RRMs are looping out RNA with RRM2 binding the 5′-motif and RRM1 the 3′-motif. (A–B) Paramagnetic relaxation enhancement (PRE) data from a spin label attached to a 4-thio-U nucleotide near the 3′-end of the ISS-N1 for (A) UP1-R140A protein bound to the modified ISS-N1 and (B) UP1-WT bound to the same modified ISS-N1. Secondary structure elements are drawn above the histograms. (C) Schematic representation of the modified ISS-N1 binding to hnRNP A1 RRMs. RRM2 is in red, and RRM1 in blue. The RNA sequence is written from the 5′-end to the 3′-end. The

*Figure 4 continued on next page*

*Figure 4 continued*

spin label is attached to a 4-thio-U located just after the RRM1 binding motif (AAGG). The spin label is represented as a yellow dot, and the PRE effect symbolized with a faint halo in yellow. (D) Surface representation of the residues in UP1-WT affected by the presence of the spin label near the 3′-end of the modified ISS-N1 RNA. The surface of hnRNP A1 RRMs is represented in red and blue for RRM2 and RRM1, respectively. Residues for which PRE data is not available due to a missing assignment in the bound form or due to severe signal overlaps are colored in gray in order to make them not appear as 'not affected'. Residues with a ratio of the intensity in the oxidized or paramagnetic state over the intensity in the reduced or diamagnetic state ($I^{para}/I^{dia}$) lower than 0.7 are colored in yellow. For facilitating the structural interpretation of the PRE data, the positions of the O4 atoms of the $U_6$ residue within the NMR bundle of hnRNP A1 RRM1 bound to 5′-UUAGGUC-3′ (this study) are shown as pink spheres. The spin label attached to the 4-thio-U in the modified ISS-N1 should sample the space around this approximate position. (*top*) front view. (*bottom*) 180 ° rotation, back view. (E) Schematic representation of hnRNP A1 RRMs binding to the modified ISS-N1-u25g RNA. RRM2 is in red, and RRM1 in blue. RRM2 binds the 5′ motif and accommodates three nucleotides (CAG), with the AG dinucleotide being stacked onto F108 and F150 of the RNP2 and RNP1 motifs, respectively. RRM1 binds the 3′ motif and accommodates four nucleotides (AAGG), with the central AG dinucleotide being stacked onto F17 and F59 of the RNP2 and RNP1 motifs, respectively. (F) Structural model of the ISS-N1-u25g bound to UP1-R140A. Modeling was performed as described in the Appendix 2. RRM2 is in red, RRM1 in blue, and the inter-RRM linker in green. The ISS-N1-u25g RNA is in yellow. Some nucleotides are labeled in order to appreciate the path of the RNA on the RRMs. Please note that the path of the RNA-spacer between the 5′ and 3′ motifs is not restrained by experimental constraints. The present structural model therefore illustrates one possible path of the ISS-N1 on the RRMs and should not be seen as the unique conformation adopted by the RNA spacer.

The following figure supplements are available for figure 4:

**Figure supplement 1.** UP1 structure and chemical shift perturbation upon binding to the ISS-N1 RNA.

**Figure supplement 2.** Summary of the intermolecular NOE between the ISS-N1-u25g RNA and the phenylalanine residues of the RNP1/RNP2 motifs of hnRNP A1 RRM1.

**Figure supplement 3.** Summary of the intermolecular NOE between the ISS-N1-u25g RNA and the phenylalanine residues of the RNP1/RNP2 motifs of hnRNP A1 RRM2.

localization and overexpression of the protein was checked by Western Blot. We observed some differences in the amounts of hnRNP A1 in the nuclear extracts (*Figure 5B*, *Figure 5—figure supplement 2*). In particular, hnRNP A1 R1r2 is consistently lower expressed. However, increasing the amount of transfected hnRNP A1 plasmid to 1 or 2 μg resulted in increased protein but had no effect on splicing (*Figure 5—figure supplement 2*). This indicates that the observed effects are independent of the dose.

Our three interface mutants INT1-3 consistently had a significant impact on *SMN1* exon 7 splicing (*Figure 5A*, lane 3–5). The interface mutants lowered exon 7 inclusion down to 25–43%, but were not capable of repressing splicing to the same extent as the wild-type protein, indicating that the relative orientation of the hnRNP A1 RRMs is essential for an optimal functioning in splicing repression.

## Both RRMs are required to bind RNA for hnRNP A1 function

We have shown in vitro that both RRMs of hnRNP A1 are bound by ISS-N1, but it is essential to know whether in cells RNA-binding of both RRMs is functionally important unlike what was recently proposed for folded RNA targets (*Morgan et al., 2015*). We therefore tested how our hnRNP A1 mutants R1r2 (F108A/F150A) and r1R2 (F17A/F59A) affect *SMN1* splicing. Having only one functional RRM for RNA binding significantly impairs exon 7 splicing repression by hnRNP A1 for both RRM2 (R1r2) and RRM1 (r1R2) (90% and 83% exon 7 inclusion, respectively) (*Figure 5A*). This effectively abolishes hnRNP A1 function. This establishes that binding of both RRMs of hnRNP A1 to RNA is required for achieving optimal repression of exon 7 inclusion.

## Interaction with both binding sites of the ISS-N1 is necessary

In our structure, the two AG dinucleotides of ISS-N1 are bound by hnRNP A1 in a singly defined orientation and both RRMs are necessary for splicing repression of *SMN1* exon 7 (*Figure 5A*). Next, we investigated if both AG sites of ISS-N1 are equally important for exon 7 inclusion by hnRNP A1 by mutating $G_{13}$, $G_{24}$ or both to adenines. Experiments were done with the *SMN2* minigene (*Hua et al., 2007*). Although in this context overexpressed hnRNP A1 can also bind to the ESS

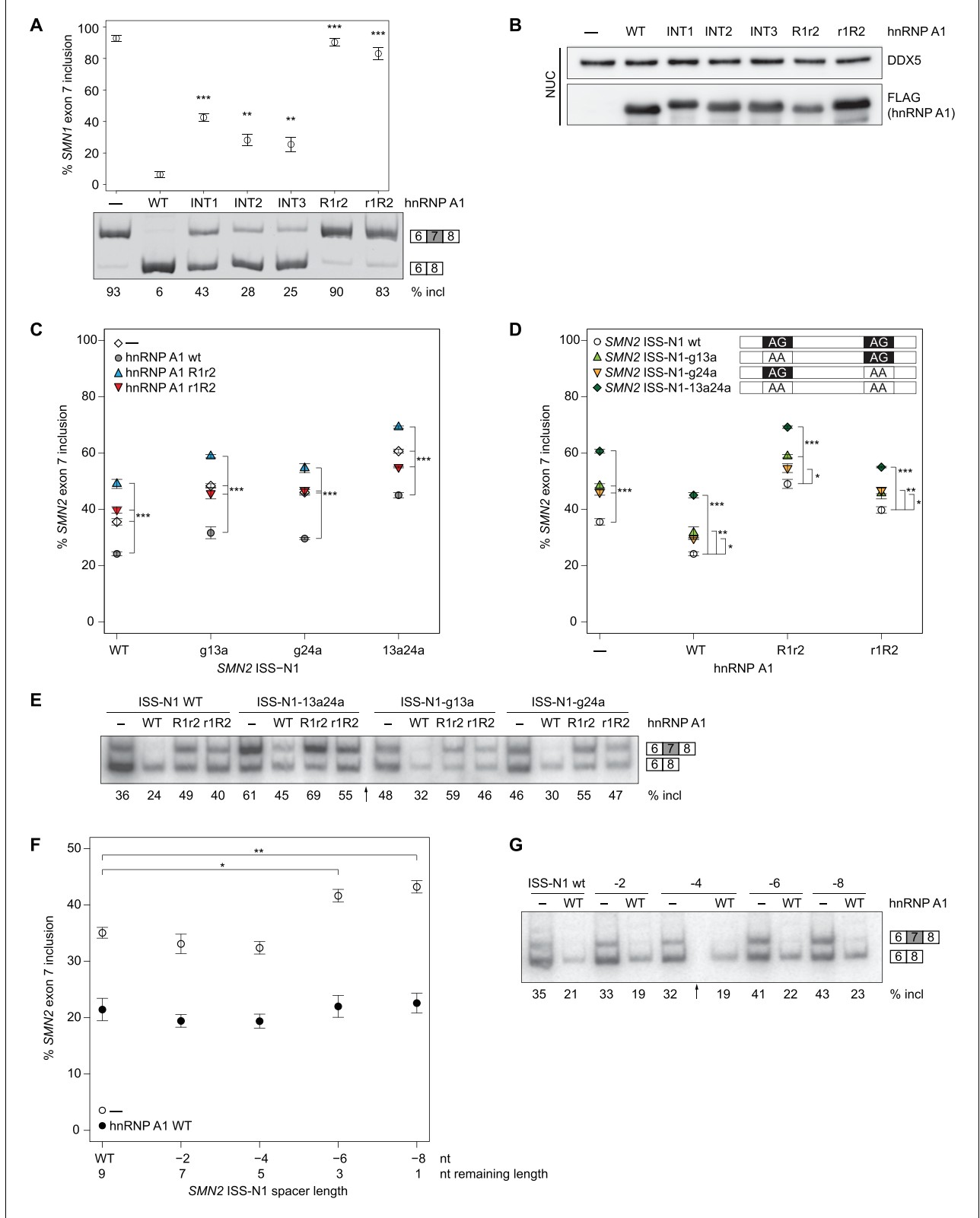

**Figure 5.** *SMN* splicing repression depends on hnRNP A1 organization and the bipartite ISS-N1 motif. (**A**) Data from alternative splicing assay in HEK293T cells. Top panel, quantification of exon 7 inclusion in *SMN1* upon overexpression of wild-type or mutant hnRNP A1. Bottom panel, PAGE with PCR products used for quantification. (**B**) Western Blot of nuclear extracts used in **A**. (**C, D**) Quantification of exon 7 inclusion in *SMN2* in relation to overexpression of wild-type or mutant hnRNP A1 (**C**) or to mutation of the hnRNP A1 binding sites in the ISS-N1 (**D**). Both figures (**C**) and (**D**) contain are

*Figure 5 continued on next page*

*Figure 5 continued*

different representations of the same data. (E) Radioautography of PAGE with PCR products used for quantification in (C, D).The separation between two individual gels is indicated by an arrow. (F) Quantification of exon 7 inclusion in *SMN2* in relation to the spacer length in-between the two hnRNP A1 binding sites within the ISS-N1. The wild-type ISS-N1 has a spacer length of 9 nt. (G) Radioautography of PAGE with PCR products used for quantification in (F). The separation between two individual gels is indicated by an arrow. Data information: All data points represent the mean of the biological replicates. Error bars correspond to the S.E.M. (B) **p=0.00113 (INT2), p=0.00360 (INT3), ***p<0.001; (C) ***p<0.001; (D) *p<0.5, **p<0.01, ***p<0.001; (F) *p=0.01176, **p=0.00287. All tested with one-way ANOVA (WT = 0), for panel (C, D) *SMN2* wt co-transfections n = 4, otherwise n = 3. The mean of the replicate is always given below the corresponding PAGE for a better readability. If not otherwise indicated 0.5 µg hnRNP A1 were co-transfected with 1 µg of the mini-gene.

The following source data and figure supplements are available for figure 5:

**Source data 1.** PAGE quantification for SMN exon 7 inclusion (%).
**Figure supplement 1.** Protein mutants disrupt the inter-RRM interface.
**Figure supplement 2.** No dose dependency of hnRNP A1 overexpression.
**Figure supplement 3.** Western blots for *SMN2* exon 7 splicing experiments in HEK293T cells.
**Figure supplement 4.** hnRNP A1 RRM mutants can not complement each other in *SMN2* splicing assay.

located in exon 7, its interaction with ISS-N1 was shown to be crucial for splicing of the exon. Indeed, the therapeutic ASO blocking the access of hnRNP A1 to ISS-N1 on *SMN2* (SPINRAZA (Nusinersen)) had a very strong effect on exon 7 inclusion (*Hua et al., 2008*). Therefore, to maximize our chance to observe an effect on splicing and better understand the mode of action of this drug, we decided to work with the *SMN2* construct. Protein overexpression levels were again assessed by Western Blot (*Figure 5—figure supplement 3*) and correspond to our observations of the hnRNP A1 co-transfections with *SMN1*.

In the absence of hnRNP A1 overexpression, we can observe 36% exon 7 inclusion for *SMN2* WT, which can be significantly lowered to 24% upon hnRNP A1 overexpression (*Figure 5C,E*). Removal of the RNA-binding ability of a single RRM impairs splicing repression (R1r2 49%, r1R2 40%, *Figure 5C,E*). However, while r1R2 has a tendency to have no impact on exon 7 inclusion compared to no hnRNP A1 overexpression, the R1r2 mutant results in an even higher exon 7 inclusion. For our experiments in the *SMN1* system the high constitutive exon 7 inclusion precludes any observation of a similar kind, although hnRNP A1 r1R2 was consistently observed to lead to a slightly lower exon 7 inclusion than hnRNP A1 R1r2. Overall, hnRNP A1 overexpression reduces exon 7 inclusion to a minimal level in both *SMN1* and S*MN2*. Additionally, the dominant negative effect observed for the RRM2 mutant in the context of *SMN2* splicing suggests that the RRMs are non-equivalent and that both RRMs of hnRNP A1 are required for *SMN* exon 7 splicing repression. Our structural data proposes the simultaneous binding of both RRMs to the ISS-N1. If binding of two molecules of hnRNP A1 to the ISS-N1 involves each an RRM (similar to [*Ding et al., 1999*; *Myers and Shamoo, 2004*], *Appendix 1—figure 1*), co-transfection of hnRNP A1 R1r2 and hnRNP A1 r1R2 could in theory rescue splicing. In support of our model, we could not observe any rescue of hnRNP A1 function when co-transfecting with the two RRM mutants (*Figure 5—figure supplement 4*).

Mutation of both AG core dinucleotides to AA (ISS-N1-13a24a) raises as expected *SMN2* exon 7 inclusion to 61%. Overexpression of hnRNP A1 can still reduce the level of exon 7 inclusion (45%) but not to the same level as in the wild-type *SMN2*-minigene (24%). Removal of either the upstream or downstream AG (ISS-N1-g13a, ISS-N1-g24a) also increases exon 7 inclusion to 48% and 46%, respectively. This can be partially suppressed to 32% and 30%, respectively, by hnRNP A1 overexpression for both ISS-N1 mutants (*Figure 5D,E*). To be sure that this observation did not result from a fortuitous enhancer element creation by our mutations, we checked our sequence with ESEfinder for known SR protein binding sites (*Cartegni et al., 2003*; *Smith et al., 2006*), which did indicate that none were introduced. Similar to the wild-type ISS-N1, hnRNP A1 r1R2 has no effect on exon 7

inclusion levels when compared to the control transfection, whereas hnRNP A1 R1r2 overexpression leads to significantly higher inclusion. This pattern holds over all three ISS-N1 mutants (*Figure 5D,E*).

The effect observed for the ISS-N1-13a24a mutant is approximately the sum of the effects of the two single-substitutions, suggesting an additive rather than a synergistic effect of the two motifs for splicing repression. The same holds true when hnRNP A1 is overexpressed, with the RNA double-mutant being less efficient for targeting a splicing repression by hnRNP A1 than the single mutants. Altogether, this shows that both RNA-binding sites of the ISS-N1 are equally important for splicing repression of exon 7.

### The spacer length does influence exon 7 inclusion

As stated before, our structural model suggests that a minimal spacer of at least 4 nucleotides is necessary to allow binding of both RRMs of a single hnRNP A1 molecule to ISS-N1. Therefore, we tested the effect of shortening the 9-nucleotide spacer by steps of 2 nt (ISS-N1 −2 to ISS-N1 −8 mutants) (*Appendix 2—table 3*).

Transfection of the *SMN2* minigene containing the 2-nt deletions showed a significant increase in exon 7 inclusion for ISS-N1 −6 and ISS-N1 −8. In contrast the ISS-N1 −2 and ISS-N1 −4 *SMN2* minigene mutants showed no aberration of the splicing pattern (*Figure 5F,G*), although this is only observed in the absence of any overexpression of hnRNP A1. In the presence of hnRNP A1 overexpression, even under a minimal one nucleotide spacer, the ISS-N1 remains a strong response element. Likely because both binding sites are preserved. In support of our model, splicing repression by hnRNP A1 via the ISS-N1 tends to be less effective when the spacer spans less than four nucleotides.

Altogether, our splicing functional data in cells are in perfect agreement with the proposal that a single hnRNP A1 molecule recognizes both AG-motifs of ISS-N1 using both RRMs (RRM2 binding upstream and RRM1 binding downstream) in a context where both RRMs interact intramolecularly as depicted in *Figure 4E,F*.

## Discussion

In this report, we investigated the RNA-binding properties of hnRNP A1 RRMs, and particularly, the RNA-binding mode of hnRNP A1 to the ISS-N1 motif that silences splicing of the *SMN* pre-mRNAs. We could show that both RRMs can bind ISS-N1 in vitro, with RRM2 binding the upstream AG motif. Structure-based mutations and splicing assays in cells are consistent with this mode of binding. Altogether, our findings have important implications for understanding the RNA-binding mode of hnRNP A1 and its mechanism of action in splicing regulation as discussed below.

### RNA-binding topologies of hnRNP A1 RRMs

In this work, we could determine the optimal recognition motif for RRM1 and RRM2 as 5′-U/$_C$AGG-3′ and 5′-U/$_C$AGN-3′, respectively. The same preference was found by Jain and co-workers using HTS-EQ on the HIV ESS3 hnRNP A1 binding site (*Jain et al., 2017*). Their consensus motif of 5′-YAG-3′ also favored a G immediately 3′ to the motif. Together this matches our findings for a strong preference of 5′-UAG-3′, which can be extended to 5′-U/$_C$AGG-3′ for a wider description of the preference. This can explain the increased predilection of hnRNP A1 for *SMN2* exon 7 caused by the C-to-U mutation at position 6 (*Kashima and Manley, 2003*).

On the basis of the crystal structure of UP1 bound to telomeric DNA repeats, a dimerization of hnRNP A1 RRMs was proposed for binding longer RNA stretches and/or to distant RNA-binding motifs on pre-mRNAs such as alternative 5′-splice sites (*Ding et al., 1999*). We could show here that hnRNP A1 RRMs can bind to RNA containing two AG-binding sites without dimerization. This simultaneous binding of the same RNA by the tandem RRMs loops out the linker RNA sequence (*Figure 4E,F*). In addition, we show that hnRNP A1 binds ISS-N1 with a directionality, RRM2 contacting the 5′ motif and RRM1 the 3′ one. The mutations that we have introduced in the protein and/or the RNA to improve the quality of the NMR data could potentially influence this mode of binding. The protein mutant UP1-R140A is however unlikely to influence UP1's topology of binding. The affected side chain does not make sequence specific contacts with the RNA and only reduces the affinity of RRM2 by 2-fold (*Table 2*). In the case of the RNA mutant ISS-N1-u25g, the mode of binding determined with the protein mutant is preserved with the wild-type protein, as revealed by the

spin-label experiments (*Figure 4B*). In addition, as far as we could judge from our NMR titrations (*Figure 3*, *Figure 3—figure supplement 2* and Appendix 2), the mode of binding determined here is rather insensitive to variations in the RNA sequence that is recognized. Indeed, several point mutations in the RNA that could potentially affect the binding of one or the other RRM did not appear to change the binding topology (*Figure 3* and *Figure 3—figure supplement 2*). This indicates that directionality is probably driven by additional contacts from the protein towards the linker RNA. Any binding motifs separated by spacers of a similar length as ISS-N1's down to four nucleotides could potentially be accommodated in the same manner based on our structural model and our splicing assays (*Figures 4* and *5*). While a shorter spacer would prevent the simultaneous binding of both RRMs, a different situation has to be considered for very long spacers. It is perfectly conceivable that hnRNP A1 RRMs could bind to such RNA-motifs simultaneously, but it is rather unlikely that the directionality observed here would be the only one occurring in such situations. This new information can be very helpful to design new ASO therapeutic oligos preventing hnRNP A1 binding to pre-mRNA such as the one developed against SMA (SPINRAZA (Nusinersen)). Our study explains why this FDA approved drug is so efficient in blocking A1 recruitment on ISS-N1 as it targets both RRM binding sites.

The question still stands on whether dimerization of UP1 monomers may be the preferred mode for binding to telomeric DNA in cells or at least in solution. Experimental evidence could so far not provide a firm answer (*Ding et al., 1999*). However, in the case of RNA stretches from natural pre-mRNAs, the mode of binding determined here for ISS-N1 is likely to be found in many regulatory binding sites of hnRNP A1. Indeed, since the spacer length gives some leeway to combine either very close or relatively distant RNA-binding motifs, a situation with optimal or sub-optimal hnRNP A1's binding motifs that associate to build a longer regulatory sequence for both RRMs of hnRNP A1 is certainly occurring many times in pre-mRNAs (*Bruun et al., 2016*). As a matter of fact, since individual RNA-binding motifs for single RRMs are short (from 3 to 4 nucleotides), it is highly probable to find two binding motifs next to each other, if one does not shorten the length of the spacer too much. CLIP data for hnRNP A1 as well as other multi-RBD containing proteins might be differentially re-examined in this light, to take into account that binding can occur to a bipartite motif spaced by at least four nucleotides. However, only further structure-function studies with other RNA-targets can reveal whether the mode of binding observed here can be generalized to other hnRNP A1 binding sites.

## Together, the two tandem RRMs form the functional RNA-binding platform of hnRNP A1

The close interaction between RRM1 and RRM2 of hnRNP A1 governs how RNA is bound and recognized by the individual RRMs. High affinity binding of RRM1, can only be reached by supplementing RRM1 with the IRL. The importance of this interplay is reflected by the strong evolutionary conservation of both RRM1 and the IRL (*Mayeda et al., 1998*). The additional contacts provided by the IRL (*Ding et al., 1999*; *Morgan et al., 2015*) help increasing the RNA-binding affinity of RRM1 by several orders of magnitude (*Table 4*). Furthermore, the presence of a guanine 3′ to the AG core motif is no longer necessary to reach nanomolar affinity.

It is the inter-RRM interaction that defines and constrains the IRL. Hence, disruption of this interface impedes hnRNP A1 function as a splicing repressor (*Figure 5A*). Comparable observations have been made upon disruption of the RRM3-RRM4 interface of PTBP1 (*Lamichhane et al., 2010*). In previous experiments, out of several chimera, only the double-RRM2 hnRNP A1 construct tested by Mayeda *et al.* was as active as hnRNP A1, most probably because RRM2 is an independent domain and has the same RNA binding properties as within UP1 or hnRNP A1 (*Table 4*) (*Mayeda et al., 1998*). However, to observe such a result it is crucial for the single RRM2 construct to contain the C-terminal helix. Constructs excluding α3 in isolated RRM2 (*Levengood et al., 2012*) or UP1 (*Shamoo et al., 1994*) or with mutations preventing the folding of α3 (M186A, *Tables 2* and *3*) perform similarly to RRM1 in isolation in terms of affinity decrease. While for many studies it can be necessary to examine the behavior of single domains in isolation, this example of the hnRNP A1 RRMs illustrates the importance of the context placement to decipher the biological function.

## Mechanism of splicing regulation by hnRNP A1

The ability by both RRMs to bind RNA is a prerequisite for hnRNP A1 mediated splicing repression (*Figure 5A,C*) (*Mayeda et al., 1994*, *1998*). Our splicing experiments in cells demonstrate that both RRMs are crucial and necessary for strong *SMN* exon 7 splicing repression.

However, the individual mutations of the two RRMs were not equivalent. We could observe a dominant negative effect only for the hnRNP A1 R1r2 variant in *SMN2* splicing but not for the r1R2 variant. A conceivable explanation could be that the different binding geometries upon RRM mutation (as illustrated in *Figure 6A*) have different downstream effects, either through interaction with endogenous hnRNP A1 or other proteins bound to the RNA.

The splicing silencer examined in this study is a powerful element that contributes to exon 7 repression of *SMN2*. We provide now evidence that loss of a single hnRNP A1 binding site in the ISS-N1 of *SMN2* is sufficient to impair exon 7 splicing suppression by hnRNP A1 to an extent where even overexpression of hnRNP A1 cannot rescue the aberrant splicing (*Figure 5D*). As anticipated, loss of both binding sites results in even higher exon 7 inclusion. Similar observations have been made by Hua and coworkers (*Hua et al., 2008*).

To a certain extent it is surprising that the loss of a single binding site within the ISS-N1 had such an impact on exon 7 repression since it requires several hnRNP A1 binding sites (*Kashima and Manley, 2003*; *Kashima et al., 2007*; *Doktor et al., 2011*; *Singh et al., 2015*). Since shortening the spacer length below four nucleotides between the two AG sites had a negative effect on exon 7 splicing repression (*Figure 5F*), and since no rescue of hnRNP A1 function was observed when co-transfecting both single RRM mutants impaired in their RNA-binding capacities (R1r2 and r1R2; *Figure 5—figure supplement 4*), we can conclude that ISS-N1 is most likely bound by a single hnRNP A1 protein in cells.

Taking into account, that hnRNP A1 can spread cooperatively with a preference to do so in the 3′-to-5′ direction (*Okunola and Krainer, 2009*), one could envisage from our data that hnRNP A1 may oligomerize initially from ISS-N1 binding. Further less specific binding can occur at the U1 snRNA binding site and extend well into exon 7, where it could join with the hnRNP A1 bound to the ESS motif at position 6 of exon 7. Accordingly, it is known that the distance of the ISS-N1 in respect to exon 7 is important and that moving it further downstream weakens its repressive character (*Singh et al., 2006*). Moreover, in a severe SMA mouse model a modified U1 snRNA targeted to bind just downstream of the ISS-N1 can rescue exon 7 splicing (*Rogalska et al., 2016*) in agreement with a competition between hnRNP A1 and U1 snRNA for binding at the very start of intron 7. Spreading in the 3′-to-5′ direction could originate from the position of RRM2 at the 5′-end of ISS-N1. Oligomerization of hnRNP A1 along the *SMN2* pre-mRNA is likely to be mediated not only by RRM-RNA binding but also by the G-domain which could interact with other hnRNP A1 molecules and RNA. Hence RRM2 could then be crucial to bring the G-rich tail near the RNA 5′-end to spread from there (*Figure 6B*).

Taken together we show that hnRNP A1 splicing repression of *SMN* strongly depends on both RRMs, which must bind RNA, and that the architecture and organization of the two RRMs of hnRNP A1 strongly influences its function. It remains to be elucidated whether cooperative hnRNP A1 oligomerization serves as a general mechanism of splicing repression and whether pre-mRNA binding will at all times occur via both RRMs.

## Material and methods

### Protein and RNA preparation for structural and ITC studies

Tandem and isolated RRMs of hnRNP A1, that is, UP1, RRM1 (2–97) and RRM2 (95–196), were expressed and purified as previously described (*Barraud and Allain, 2013*). Mutagenesis experiments were performed using the Quikchange Kit (Stratagene) following the manufacturer's instructions. All mutant proteins were checked to be properly folded by running ($^{15}$N,$^{1}$H)-HSQC or ($^{1}$H)−1D NMR spectra. Short RNA oligonucleotides were purchased from Dharmacon, deprotected according to the manufacturer's instructions, lyophilized and resuspended in the NMR buffer (NaPi pH 6.5 10 mM, DTT 1 mM). NMR samples of protein–RNA complexes were prepared at 0.8–1.0 mM at a ratio of 1:1 in a sample volume of 250 μL. Long RNA oligonucleotides, that is, ISS-N1 wild-type and mutants, were prepared by *in vitro* transcription with T7 polymerase (RNA sequence 5′-GGA

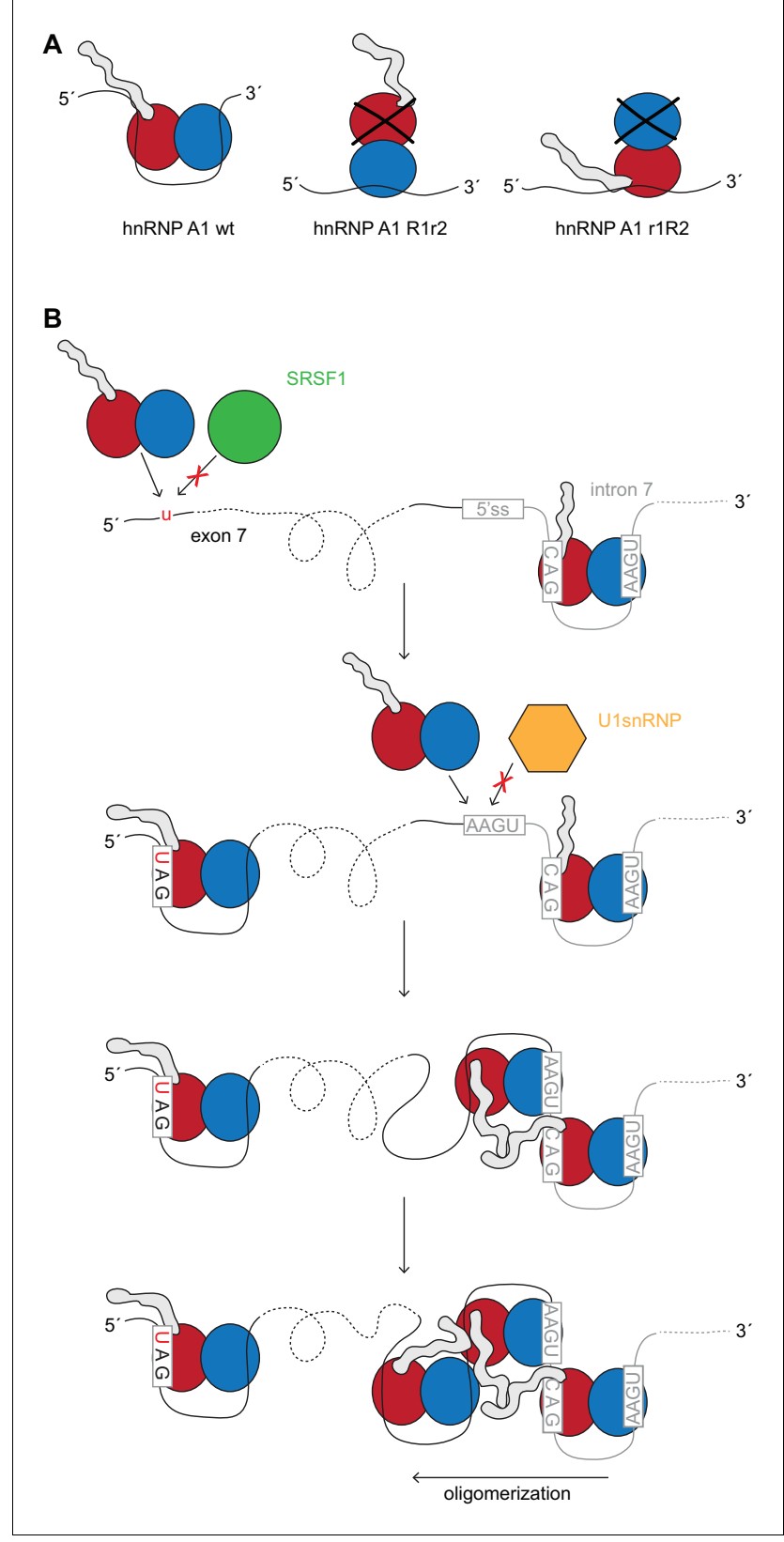

**Figure 6.** HnRNP A1 splicing repression and its dependency. (**A**) Model of hnRNP A1 mutants R1r2 and r1R2. Owing to the protein's topology the glycine-rich tail will be closer or further away from the pre-mRNA. Our data suggests that splicing repression depends on RRM2 RNA binding which might bring the glycine-rich domain

*Figure 6 continued on next page*

*Figure 6 continued*

closer to the RNA and/or other protein factors. (B) Model of hnRNP A1 assembly on the *SMN2* exon 7 and intron 7. Starting at the ISS-N1 hnRNP A1 could oligomerize along the 3′-to-5′ direction. The G-rich tail can interact with other hnRNP A1 molecules. Binding sites are given by the U1 snRNA binding site and the $C_6$-to-T mutation at the start of exon 7.

CCAGCAUUAUGAAAGUGA-3′; with adding an initial GGA upstream the ISS-N1 sequence for efficient transcriptions). RNA was purified by anion-exchange high-pressure liquid chromatography under denaturing conditions, as previously described (*Barraud et al., 2012*).

## NMR spectroscopy and resonance assignments

All NMR measurements were conducted in the same buffer (NaPi pH 6.5 10 mM, DTT 1 mM) at a temperature of 303 K. NMR spectra were measured on Bruker AVIII-500 MHz, AVIII-600 MHz, Avance-600 MHz, AVIII-700 MHz, AVIII-750 MHz and Avance-900 MHz spectrometers. Except for the 750 MHz, all spectrometers were equipped with cryoprobes. The data were processed using TOP-SPIN 3.x (Bruker) and analyzed with Sparky (www.cgl.ucsf.edu/home/sparky/).

Protein resonances were assigned with 2D ($^1$H,$^{15}$N)-HSQC, 2D ($^1$H,$^{13}$C)-HSQC, 3D HNCA, 3D HNCACB, 3D CBCA(CO)NH, 3D HNCO, 3D HN(CA)CO, 3D [$^{13}$C; $^{15}$N; $^1$H] HCC(CO)NH-TOCSY, 3D [$^1$H; $^{15}$N; $^1$H] HCC(CO)NH-TOCSY, 3D NOESY-($^1$H,$^{15}$N)-HSQC and two 3D NOESY-($^1$H,$^{13}$C)-HSQC optimized for the observation of protons attached to aliphatic carbons or to aromatic carbons. In addition, the assignment of aromatic protons was conducted using 2D ($^1$H,$^1$H)-TOCSY and 2D ($^1$H,$^1$H)-NOESY measured in $D_2O$.

To assign the resonances of unlabeled RNAs, we used 2D ($^1$H,$^1$H)-TOCSY, 2D ($^1$H,$^1$H)-NOESY, and 2D $^{13}$C F1-filtered F2-filtered ($^1$H,$^1$H)-NOESY (*Peterson et al., 2004*). Protein–RNA intermolecular NOEs were obtained by 2D ($^1$H,$^1$H)-NOESY, 2D $^{13}$C F2-filtered ($^1$H,$^1$H)-NOESY (*Zwahlen et al., 1997*) and 3D $^{13}$C F1-edited, F3-filtered [$^1$H; $^{13}$C; $^1$H] NOESY-HSQC spectrum (*Zwahlen et al., 1997*), all recorded in $D_2O$. We recorded NOESY spectra with a mixing time of 120–150 ms. 2D NOESYs were also measured at pH 5.5 and 280 K for optimizing the observation of RNA iminos. For the NMR dynamics study, $^{15}$N T1 and T2 measurements were recorded at 303 K at a $^1$H frequency of 600 MHz with established methods (*Kay et al., 1989*; *Skelton et al., 1993*) as previously described (*Barraud and Allain, 2013*).

## Structure calculation of the protein-RNA complexes

Automated NOE cross-peak assignments (*Herrmann et al., 2002a*) and structure calculations with torsion-angle dynamics (*Güntert et al., 1997*) were performed with the macro 'noeassign' of the software package CYANA 3.0 (*Güntert, 2004*). Unassigned peak lists of the protein 3D NOESY spectra were generated as input with the program ATNOS (*Herrmann et al., 2002b*) and manually cleaned to remove artifact peaks. Intramolecular RNA and protein-RNA intermolecular NOEs were manually assigned but were not manually converted into distance constraints. Instead, peak intensities were automatically calibrated and converted to distance constraints by CYANA. The input also contained hydrogen-bond restraints determined as follows: intramolecular hydrogen-bonded amides were identified as slowly exchanging protons in presence of $D_2O$; their hydrogen-bond acceptors were identified from preliminary structures as well as from analysis of the characteristic NOE pattern found in α-helices and β-sheets. For protein-RNA intermolecular hydrogen-bonds, donors and acceptors were identified from preliminary structures and confirmed by large NH or CO chemical shift displacement upon RNA binding. In addition, dihedral δ angles of sugars were restrained to the C2′-endo conformation for nucleotides with a strong H1′-H2′ cross-peak in the TOCSY spectrum. We calculated 100 independent structures that we refined in a water shell with the program CNS 1.3 (*Brünger et al., 1998*; *Brunger, 2007*) including distance restraints from NOE data, hydrogen-bond restraints, and dihedral angles of sugars, as previously described (*Barraud et al., 2011*). Twenty structures were selected based on their energy and agreement with the NOE data, and were analyzed with PROCHECK-NMR (*Laskowski et al., 1996*) and the iCING web server (*Doreleijers et al., 2012*). Overall structural statistics of the final water-refined structures are shown in *Table 1*. Structures were visualized and figures were prepared with the program PYMOL (http://www.pymol.org).

## Spin-labeling and paramagnetic relaxation enhancement

A modified RNA (5´-ACCAGCAUUAUGAAAGG(4-thio-U)A-3´) was purchased from Dharmacon and deprotected according to the manufacturer's instructions. The 3-(2-iodoacetamido)proxyl (IA-proxyl) spin-label was dissolved in 100% methanol to obtain a 100 mM concentration. The modified RNA was resuspended into a reaction buffer containing 100 mM potassium phosphate pH 8.0, 100 µM s$^4$U-modified RNA and 10 mM IA-proxyl spin label (*Duss et al., 2014*). The reaction mix was incubated for 24 hr at 25°C in the dark with agitation. Reaction progress was followed by checking the s$^4$U absorbance at 320 nm (*Ramos and Varani, 1998*). The spin-labeled RNA was purified with two successive NAP-10 columns (GE healthcare). Protein-RNA complexes at 0.2 mM were formed at a 1:1 ratio and ($^{15}$N,$^1$H)-HSQC spectra were measured before (paramagnetic sample) and after reduction of the spin-label with the addition of 2 mM ascorbic acid (diamagnetic sample). Backbone chemical shift assignment of the complexes was inferred from the assignment of the UP1-R140A:ISS-N1-u25g complex. Residues with a ratio of the intensity in the oxidized or paramagnetic state over the intensity in the reduced or diamagnetic state ($I^{para}/I^{dia}$) lower than 0.7 were considered as affected by the presence of the spin label.

## Modeling of the UP1-R140A:ISS-N1-u25g complex

Protein resonances in the complex were assigned using standard methods. The RNA anomeric and aromatic proton signals were assigned using 2D TOCSY and various 2D NOESY spectra (See Appendix 2 for details). Next, using a filtered NOESY experiments measured on a $^{13}$C/$^{15}$N-labeled protein in complex with an unlabeled RNA, we could determine the unambiguous intermolecular NOE for the assigned nucleotides (see *Appendix 2—table 2* for a summary of the assigned protein RNA intermolecular NOE). A portion of the half-filtered 2D-NOESY is presented on *Figure 4—figure supplements 2* and *3*, such as to summarize the intermolecular NOEs from the anomeric and aromatic protons of the RNA towards the aromatic protons of the phenylalanine residues from the RNP motifs (i.e. F17/F57/F59 for RRM1 and F108/F148/F150 for RRM2).

Model structures of the UP1-R140A:ISS-N1-u25g complex (*Figure 4F*) were obtained by a simulated annealing protocol with the software package CYANA 3.0 (See Appendix 2 for details) (*Güntert, 2004*). We calculated 100 independent structures that we refined in a water shell with the program CNS 1.3 (*Brünger et al., 1998*; *Brunger, 2007*). Models were selected based on their energy of interaction at the protein-RNA interface (including the Van der Waals and the electrostatic terms of the CNS energy function restricted to the inter-molecular interactions). The lowest energy model was chosen for figure preparation (see *Figure 4F*).

## Isothermal titration calorimetry

Isothermal titration calorimetry (ITC) experiments were performed on a VP-ITC instrument (MicroCal Inc, Wolverton Mill, UK). Optical density absorbance at 280 and 260 nm was used to determine the concentrations of proteins and RNAs, respectively. Samples were prepared in NMR buffer (replacing 1 mM DTT with 10 mM 2-mercaptoethanol). The measurements were performed using the protein as titrant and the RNA as titrate. All measurements were taken at a cell temperature of 30°C over (40-)44 injections, each 6 µL, with a spacing time of 300 s and a filter period of 2 s. The reference power was set to 10 µCal/s and the initial delay to 60 s. The data was analyzed using the software Origin version 7.0 (MicroCal Inc) and a 1:1 binding model using nonlinear least-squares fitting. For measurements with isolated RRM1 or RRM2 (WT and mutants) the baseline was set manually for all experiments prior to integration, and the data was corrected for nonspecific heats. For measurements with the UP1 mutants no baseline correction was applied and the data was corrected for dilution effects by subtracting the integrated heats of a control experiment of protein titration into buffer prior to fitting.

The output of the ITC measurements can be found in *Table 3—source data 1* and *Table 4—source data 1*, *Table 4—source data 2*.

## Cell culture and plasmids

HEK293T (human embryonic kidney) cells were obtained from the European Collection of Cell Cultures (ECACC No. 85120602) and were cultured in Dulbecco's modified Eagle's medium (DMEM) supplemented with 10% fetal bovine albumin (FBS) and 1% streptomycin/penicillin. The cells were

checked for mycoplasma contamination with GATC's MYCOPLASMACHECK and found to be clean. The pCI-SMN1 and pCI-SMN2 plasmids containing the *SMN1* or *SMN2* minigene, respectively, were previously described (*Hua et al., 2007*). For hnRNP A1 overexpression we used the pCGT7 hnRNP A1-A plasmid (*Cáceres et al., 1997*) but with the T7 tag substituted by a FLAG tag (pCGFLAG hnRNP A1). The hnRNP A1 r1R2 (F17A/F59A), R1r2 (F108A/F150A), INT1 (R75A/D157K), INT2 (D155R/D157K) and INT3 (D155R/D157K/I164A) mutants as well as the SMN1 and SMN2 ISS-N1 mutants were generated by side-directed mutagenesis using specific primers (*Liu and Naismith, 2008*).

### In vivo splicing assay

One microgram of pCI-SMN1 and SMN2 (wild-type (WT) or mutant) was co-transfected with 0.5 μg or 1.0 μg or 2.0 μg of pCGFLAG hnRNP A1 (WT or mutant), respectively, in HEK293T cells counted and seeded 24 hr in advance at 400'000 cells/well, plated in six-well plates using the Lipofectamine 2000 Reagent (Lifetechnologies, CA, United States) according to the manufacturers protocol. After 48 hr total RNA was extracted and 1 μg was used for reverse transcription using Oligo(dT)$_{15}$ primer and M-MLV Reverse Transcriptase RNase (H-) (Promega). 10% of the resulting cDNA was then used for semiquantitative PCR using a vector specific forward primer (pCI-fwd 5′-GGTGTCCACTCCCAGTTCAA-3′) and a *SMN1* specific reverse primer (SMN1rev 5′-AGCCTTATGCAGTTGCTCTC-3′) or a *SMN2* specific reverse primer (SMN2rev 5′-GCCTCACCACCGTGCTGG-3), respectively. For *SMN2* samples, the PCR was run with $^{32}$P 5-labeled SMN2rev primer. The bands corresponding to the product of the splicing reaction were quantified using AlphaView (proteinsimple, San Jose, California) from radioautography of a 4% polyacrylamide gel for *SMN2* samples and by staining the gel with GelRed (Biotium) for *SMN1* samples. The ratio of each isoform was normalized to the sum of isoforms. Experiments were repeated three times independently allowing for the calculation of the mean and standard error of the mean for each assay.

### Western blotting

Fractionation of nuclear and cytoplasmic fractions was done with the NE-PER kit (Thermo Scientific). Protein samples were separated by 15% SDS-PAGE and then electroblotted onto nitrocellulose membranes (Amersham Protran 0.2 NC, GE Healthcare), blocked with 5% non-fat dry milk in Tris-buffered saline +0.1% Tween (TBS-T). Probing of the blots was done with monoclonal anti-FLAG (F7425, Sigma) in TBS-T and 5% non-fat dry milk, or anti-DDX5 (D15E10, Cell Signaling) in TBS-T, washed three times in TBS-T and 5% non-fat dry milk, followed by incubation with horseradish-peroxidase-conjugated anti-mouse or anti-rabbit secondary antibodies (Sigma) for 2 hr in TBS-T and 5% non-fat dry milk. Protein signals were detected with chemiluminescence imaging (Amersham Imager 600RGB).

### Accession codes

The chemical shifts of hnRNP A1 RRMs in complex with RNA have been deposited in the Biological Magnetic Resonance Bank under accession numbers 34079 and 34080 for RRM1 and RRM2, respectively. The coordinates of the structures have been deposited in the Protein Data Bank under accession codes 5MPG and 5MPL for RRM1 and RRM2, respectively.

## Acknowledgement

We thank A Krainer for the initial hnRNP A1 plasmid, D Schümperli for the initial *SMN1* plasmid, and S Stamm for the initial *SMN2* plasmid; P Lukavsky for providing specifically deuterated nucleotides; S Jayne and C von Schrötter for initial in vivo splicing assays; J Boudet for discussion of the ITC data; O Voinnet for access to lab infrastructure and S Oberlin for discussion/help with Western Blots; F Damberger, G Wider, T Suter-Stahel, M Schubert and C Maris for ensuring the best performance of the NMR infrastructure.

## Additional information

### Funding

| Funder | Grant reference number | Author |
|---|---|---|
| Centre National de la Recherche Scientifique | | Pierre Barraud |
| ETH Fellowship Program | Post-doc fellowship | Pierre Barraud |
| Novartis Foundation | Post-doc fellowship | Pierre Barraud |
| Cure SMA | | Antoine Cléry<br>Frédéric Hai-Trieu Allain |
| Fondation Suisse de Recherche sur les Maladies Musculaires | | Antoine Cléry<br>Frédéric Hai-Trieu Allain |
| Eidgenössische Technische Hochschule Zürich | | Frédéric Hai-Trieu Allain |
| Schweizerischer Nationalfonds zur Förderung der Wissenschaftlichen Forschung | | Frédéric Hai-Trieu Allain |
| Schweizerischer Nationalfonds zur Förderung der Wissenschaftlichen Forschung | | Frédéric Hai-Trieu Allain |
| SMA Europe | | Frédéric Hai-Trieu Allain |
| NCCR Structural Biology | | Frédéric Hai-Trieu Allain |
| NCCR RNA and Disease | | Frédéric Hai-Trieu Allain |

The funders had no role in study design, data collection and interpretation, or the decision to submit the work for publication.

### Author contributions

IB, Conceptualization, Formal analysis, Validation, Investigation, Visualization, Writing—original draft, Writing—review and editing; PB, Conceptualization, Formal analysis, Supervision, Funding acquisition, Validation, Investigation, Visualization, Writing—original draft, Writing—review and editing; AM, Formal analysis, Validation, Investigation, Writing—review and editing; AC, Supervision, Validation, Investigation, Writing—review and editing; FH-TA, Conceptualization, Formal analysis, Supervision, Funding acquisition, Validation, Investigation, Visualization, Writing—original draft, Project administration, Writing—review and editing

### Author ORCIDs

Irene Beusch, http://orcid.org/0000-0001-7758-4348

Pierre Barraud, http://orcid.org/0000-0003-4460-8360

Frédéric Hai-Trieu Allain, http://orcid.org/0000-0002-2131-6237

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

## Appendix 1

## Topology and RNA-binding preferences of hnRNP A1 RRMs

Structural insights into the RNA-binding mode of hnRNP A1 were mostly derived from the structure of UP1 bound to single-stranded telomeric DNA repeats (*Ding et al., 1999*). In this structure, two symmetry-related molecules of UP1 interact to form a dimer that binds two strands of DNA in an anti-parallel manner, each strand extending across the dimer interface (*Appendix 1—figure 1*). This peculiar topology could be the result of crystal packing forces or might only be relevant for the binding of hnRNP A1 to telomeric DNA repeats.

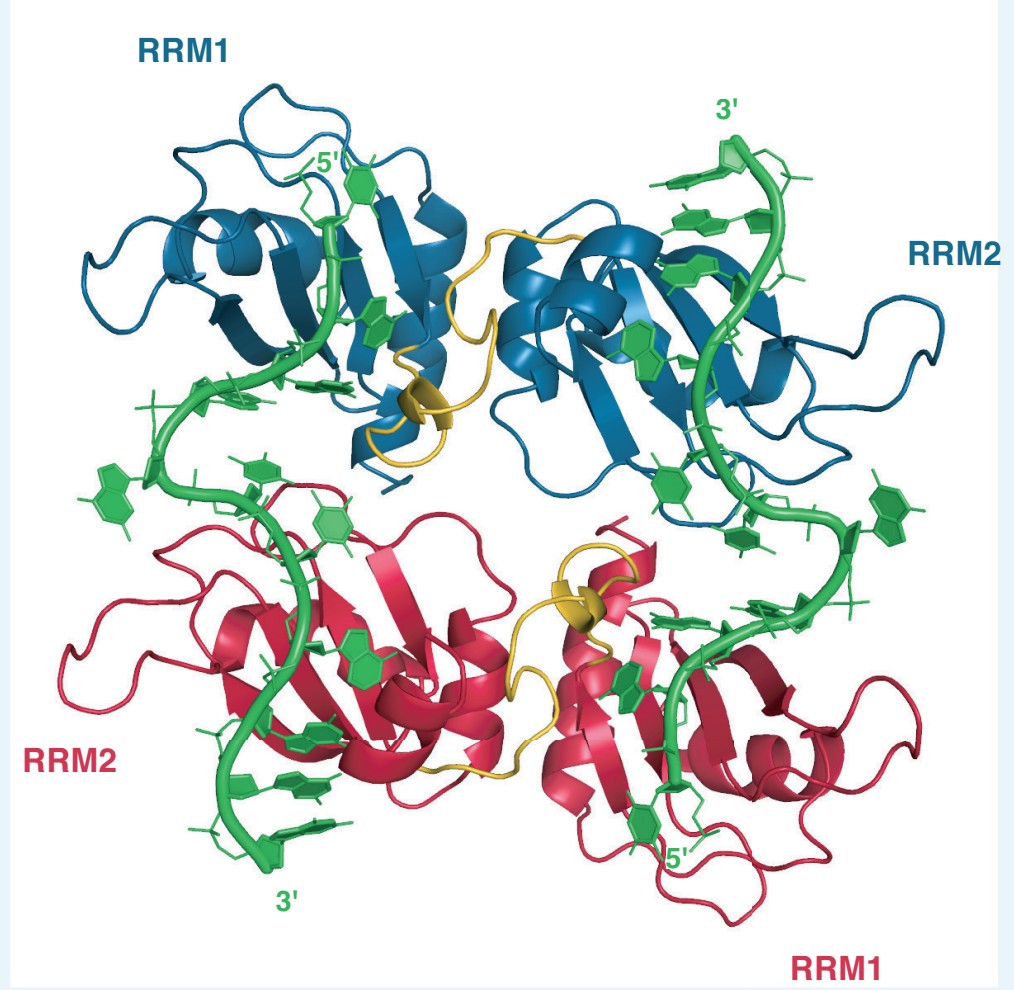

**Appendix 1—figure 1.** Crystallographic structure of hnRNP A1 RRMs (UP1) bound to telomeric DNA repeats. In this structure, one single-stranded DNA molecule consisting of 2 repeats of the TTAGGG motif is bound by two distinct UP1 molecules (one in red, one in blue) that are related by a crystallographic symmetry (C2 axis pointing perpendicular to the plan of the figure). RRM1 binds the 5′ end of the DNA and RRM2 the 3′ end. The inter-RRM linker (residues 89 to 105) is in yellow. PDB ID code 2UP1 (*Ding et al., 1999*).

In order to better characterize the differences in terms of specificities of each RRM, we decided to study each hnRNP A1 RRM in isolation. We thus performed NMR titrations of the

isolated RRMs with several short RNA ranging from 6 to 8 nucleotides, each containing a core AG dinucleotide central for hnRNP A1 binding (*Appendix 1—table 1*). Interestingly, among the different RNA sequences that we tested, we could find for each RRM one RNA sequence where complex formation is in the slow exchange regime, indicative of a strong binding.

**Appendix 1—table 1.** Sequences of the RNA oligonucleotides for which NMR titrations with either RRM1 or RRM2 have been performed and the corresponding NMR exchange regime.

| RNAs tested with RRM1 | | | RNAs tested with RRM2 | | |
|---|---|---|---|---|---|
| Sequence | Number of nucl. | Exchange regime | Sequence | Number of nucl. | Exchange regime |
| UUAGGUC | 7 nt | slow | UUAGGUC | 7 nt | fast |
| UUAGGGA | 7 nt | fast | UUAGGGA | 7 nt | fast |
| CAGCAU | 6 nt | fast | CAGCAU | 6 nt | fast |
| UGAAAGU | 7 nt | fast | UGAAAGU | 7 nt | fast |
| UCAGGUC | 7 nt | fast | UCAGGUC | 7 nt | fast |
| UCAGUU | 6 nt | fast | **UCAGUU** | **6 nt** | **slow** |
| UUUUAGGU | 8 nt | intermediate | UCAGGG | 6 nt | fast |
| CGUAGGU | 7 nt | intermediate | UUAGUU | 6 nt | intermediate |
| UAGGUC | 6 nt | fast | UAGGUC | 6 nt | fast |

## Appendix 2

# Investigating the topology of RNA binding onto hnRNP A1 RRMs

We therefore introduced guanines downstream of the AG core dinucleotide to increase the affinity (*Table 4*), creating the RNA mutants ISS-N1-c14g, ISS-N1-u25g, and ISS-N1-14g25g (*Figure 3—figure supplement 1*). Single guanine introduction resulted in spectra containing resonances with very broad line width and still many resonances disappearing (*Figure 3—figure supplement 2B,C*). Titration with ISS-N1-14g25g still resulted in broad line width, but fewer resonances disappeared compared to the wild-type sequence (*Figure 3—figure supplement 2D*). Although promising, this first approach did not achieve sufficiently narrow lines. Since it is not necessarily the complex with the best affinity that will give the best NMR signals, we then decided to take the opposite approach of weakening the overall affinity of the complex with the aim of reaching the fast exchange regime with improved NMR signals. We thus titrated UP1-R140A with the wild-type ISS-N1 (*Figure 3A,E*). The spectrum had improved quality for some regions, but also broader line-width in others. We next decided to combine our two approaches and obtained spectra of much improved quality with overall narrow line-widths and many recovered signals using UP1-R140A bound to ISS-N1-14g25g (*Figure 3B,F*). Interestingly, the titrations with the singly mutated RNAs gave also spectra of improved quality. Whereas the ISS-N1-u25g showed overall approximately identical chemical shifts as the double RNA mutant, the ISS-N1-c14g showed some localized differences (*Figure 3C,D,G,H*). More importantly, in the case of ISS-N1-14g25g and ISS-N1-u25g, the pattern of signals in the bound form in this particular region resembles those of the isolated RRM1 bound to 5´-UUAGGUC-3´ RNA (*Figure 1—figure supplement 1A*).

In detail, the residues involved correspond to Ile18, Val45, Met46 and Ala89 in RRM1, among which Val45 and Met46 ($\beta$2 strand) interact with the guanosine 3´ to the AG core (*Figure 1—figure supplement 1A*, and *Figure 3F,H*). Our interpretation is therefore the following: since the pattern of the region at 8–9 ppm x 124–130 ppm is similar for the isolated RRM1 bound to 5´-UUAGGUC-3´, and for UP1 bound to ISS-N1-14g25g and to ISS-N1-u25g, we conclude that in these cases RRM1 binds a motif with a G following the central AG core. Correlatively, in the case where the pattern is different, namely for ISS-N1-c14g, we conclude that RRM1 binds a motif with something else than a G following the central AG core. This strongly indicates that RRM1 binds to the 3´-motif of ISS-N1 irrespective of the point mutations introduced in the RNA and therefore RRM2 binds the 5´-motif.

# Structural investigation of ISS-N1-u25g bound to UP1-R140A

Protein resonances in the complex were assigned using 2D ($^1$H,$^{15}$N)-HSQC, 2D ($^1$H,$^{13}$C)-HSQC, 3D HNCA, 3D CBCA(CO)NH, 3D HNCO, 3D [$^{13}$C; $^{15}$N; $^1$H] HCC(CO)NH-TOCSY, 3D [$^1$H; $^{15}$N; $^1$H] HCC(CO)NH-TOCSY, 3D NOESY-($^1$H,$^{15}$N)-HSQC and two 3D NOESY-($^1$H,$^{13}$C)-HSQC optimized for the observation of protons attached to aliphatic carbons or to aromatic carbons. In addition, the assignment of aromatic protons was conducted using 2D ($^1$H,$^1$H)-TOCSY and 2D ($^1$H,$^1$H)-NOESY measured in D$_2$O.

We could reach 92% of proton assignments over the segment 10–189, that is, the structured part of UP1 and could calculate the structure of the RNA-bound form of UP1 with 4261 distance restraints with a backbone r.m.s.d of 1.1 ± 0.4 Å (*Figure 4—figure supplement 1A*).

The RNA anomeric and aromatic proton signals were assigned using 2D ($^1$H,$^1$H)-TOCSY, 2D ($^1$H,$^1$H)-NOESY, 2D $^{13}$C F1-filtered F2-filtered ($^1$H,$^1$H)-NOESY (*Peterson et al., 2004*) and

2D $^{13}$C F2-filtered ($^1$H,$^1$H)-NOESY (*Zwahlen et al., 1997*) all measured in D$_2$O at 900 MHz. In addition, a 2D ($^1$H,$^1$H)-NOESY spectrum was measured on a sample with an RNA deuterated on specific positions, that is, H8 of purines and H5 of pyrimidines. This simplified the spectrum and helped identifying H2 of adenines (*Duss et al., 2012*).

Although the RNA assignment was incomplete due to severe resonance overlaps, we could confidently assign the RNA around the two AG dinucleotides present in the ISS-N1 sequence (*Appendix 2—table 1*).

**Appendix 2—table 1.** Summary of the assigned proton chemical shifts of the ISS-N1-u25g RNA bound to UP1-R140A protein variant (at 303 K). Nucleotides with partial or complete assignment are in bold letters in the following sequence: 5´-GGAC**CAG**CAUUAUG**AAAGGGA**-3´.

| Nucleotide | Atom | Chemical shift (ppm) |
|---|---|---|
| CYT11 | H1´<br>H5<br>H6 | 5.74<br>5.87<br>7.78 |
| ADE12 | H1´<br>H2<br>H8 | 6.02<br>8.04<br>8.19 |
| GUA13 | H1´<br>H8 | 5.78<br>7.69 |
| ADE21 | H1´<br>H2<br>H8 | 5.87<br>8.04<br>8.28 |
| ADE22 | H1´<br>H2<br>H8 | 6.15<br>8.03<br>8.47 |
| ADE23 | H1´<br>H2<br>H5´/H5´´<br>H8´ | 6.08<br>8.42<br>2.92/3.68<br>8.18 |
| GUA24 | H1´<br>H2´<br>H3´<br>H4´<br>H8 | 6.42<br>5.86<br>5.09<br>4.59<br>8.78 |
| GUA25 | H1´<br>H8 | 5.69<br>7.95 |
| GUA26 | H1´<br>H8 | 5.66<br>7.41 |
| ADE27 | H1´<br>H2<br>H8 | 5.90<br>8.02<br>8.14 |

Next, using a half-filtered 2D NOESY and a filtered-edited 3D NOESY measured on a $^{13}$C/$^{15}$N-labeled protein in complex with an unlabelled RNA, we could determine the unambiguous intermolecular NOE for the assigned nucleotides.

Overall we observed 87 unambiguous intermolecular NOEs around the two core AGs (see *Appendix 2—table 2* for a summary of the assigned protein RNA intermolecular NOE). A portion of the half-filtered 2D-NOESY is presented on *Figure 4—figure supplements 2* and *3*, such as to summarize the intermolecular NOEs from the anomeric and aromatic protons of the RNA towards the aromatic protons of the phenylalanine residues from the RNP motifs (i.e. F17/F57/F59 for RRM1 and F108/F148/F150 for RRM2).

**Appendix 2—table 2.** Summary of the assigned intermolecular-NOE between the ISS-N1-u25g RNA and the UP1-R140A protein variant. These NOEs were observed in the 2D $^{13}$C F2-filtered ($^{1}$H,$^{1}$H)-NOESY or the 3D $^{13}$C F1-edited, F3-filtered [$^{1}$H; $^{13}$C; $^{1}$H] NOESY-HSQC or in both spectra. The intensity of the reported intermolecular-NOE peaks are classified regarding their signal/noise ratio as follows: - weak: 7 ≤ S/N <12 - middle: 12 ≤ S/N <25 - strong: 25 ≤ S/N <45 - very strong: 45 ≤ S/N.

| Nucleotide | Atom | Protein residue and atom | Intensity of inter-NOE |
|---|---|---|---|
| CYT11 | H1′ | F108QE | weak |
| | | F108HZ | weak |
| | | F148QD | middle |
| | H5 | E176HB2 | middle |
| | | E176HB3 | middle |
| | | E176QG | strong |
| | | R178QD | middle |
| ADE12 | H1′ | F108QE | weak |
| | | M137QE | middle |
| | | F148QD | middle |
| | | F148QE | middle |
| | | F148HZ | middle |
| | | F150QD | middle |
| | | F150QE | middle |
| | | A180QB | middle |
| | | L181QD1 | weak |
| | | L181QD2 | weak |
| | | K183QB | weak |
| | | K183QG | weak |
| | | K183QE | middle |
| | H2 | R178QD | strong |
| | | A180QB | strong |
| | | L181QD1 | strong |
| | | L181QD2 | very strong |
| | H8 | F108QE | weak |
| | | E176HB2 | middle |
| | | E176HB3 | middle |
| | | E176QG | strong |
| | | R178QD | strong |
| | | A180QB | strong |
| | | L181QD1 | middle |
| | | L181QD2 | very strong |
| GUA13 | H1′ | M137QE | very strong |
| | | F148QE | weak |
| | | F150QE | weak |
| | H8 | K106QE | weak |
| | | M137HG2 | middle |
| | | M137QE | strong |
| | | F148QE | weak |
| | | F150QE | weak |
| | | K183QE | middle |
| ADE22 | H1′ | F17QE | very strong |
| | | G19HA3 | weak |
| | | F57HB2 | weak |
| | | F57HB3 | weak |
| | | F57QD | strong |
| | H8 | F17QE | weak |

*Appendix 2—table 2 continued on next page*

*Appendix 2—table 2 continued*

| Nucleotide | Atom | Protein residue and atom | Intensity of inter-NOE |
|---|---|---|---|
| ADE23 | H1′ | F17QE | middle |
| | | F17HZ | middle |
| | | M46QE | middle |
| | | F57HZ | middle |
| | | F59QE | middle |
| | | F59HZ | middle |
| | H2 | F17HZ | strong |
| | | M46QE | middle |
| | | F59HB2 | middle |
| | | F59HB3 | middle |
| | | F59QD | strong |
| | | F59QE | strong |
| | | A89HA | middle |
| | | A89QB | very strong |
| | | V90HB | middle |
| | H8 | F17QD | middle |
| | | F17QE | middle |
| GUA24 | H1′ | M46QE | very strong |
| | | F59QE | weak |
| | H2′ | M46QE | very strong |
| | | F59QE | middle |
| | | F59HZ | middle |
| | H3′ | M46QE | very strong |
| | | F57HZ | middle |
| | | F59QD | middle |
| | | F59HZ | middle |
| | H4′ | V44QG1 | middle |
| | | M46QE | very strong |
| | H8 | M46QE | very strong |
| | | F57QE | middle |
| | | F59QD | middle |
| | | F59QE | middle |
| | | F59HZ | middle |
| GUA25 | H1′ | V44HB | middle |
| | | V44QG1 | very strong |
| | | V44QG2 | very strong |
| | | M46QE | very strong |
| | | F59QD | weak |
| | | F59HZ | middle |
| | H8 | M46QE | very strong |

## Modeling of the UP1-R140A:ISS-N1-u25g complex

Model structures of the UP1-R140A:ISS-N1-u25g complex were obtained by a simulated annealing protocol with the software package CYANA 3.0 (*Güntert, 2004*). The constraints include 4261 intra-protein NOE (*Figure 4—figure supplement 1A*) and 87 inter-molecular NOE (see *Appendix 2—table 2*). The input also contained hydrogen-bond restraints identical to the one identified for the single-RRM complexes. We calculated 100 independent structures that we refined in a water shell with the program CNS 1.3 (*Brünger et al., 1998*; *Brunger, 2007*). Models were selected based on their energy of interaction at the protein-RNA interface (including the Van der Waals and the electrostatic terms of the CNS energy function restricted to the inter-molecular interactions). The lowest energy model was chosen for figure preparation (see *Figure 4F*).

## Important remark

It is important to note that this model only illustrates one possible path of the ISS-RNA from RRM2 towards RRM1. The RNA-spacer separating the two RRM-bound motifs of the ISS-N1 is not restricted in this modelling procedure, and therefore can adopt a large number of different conformations. Nevertheless, this model illustrates that the RNA-spacer is long enough for hnRNP A1 RRMs to loop out RNA and bind simultaneously to the 5´- and 3´-motifs.

## Influence of the RNA spacer

In order to evaluate the influence of the RNA-spacer on the topology of binding observed here, namely RRM1 binding the 3'-motif and RRM2 the 5'-motif, we conducted several structure calculations with CYANA using the same protocol used for the UP1-R140A:ISS-N1-u25g complex. In each of these calculations, we progressively shortened the length of the RNA spacer from 9 nt to 3 nt (see *Appendix 2—table 3*). As mentioned before, we define the term 'RNA spacer' as the number of nucleotides in between the two AG core binding sites. Several lengths were tested, namely 9 nt (the original sequence), 7 nt, 5 nt, 4 nt and 3 nt. The structure calculation with the 3-nucleotide long spacer resulted in a distorted geometry for the protein-RNA complex with large violations, reflecting the fact that the two RRMs cannot bind simultaneously to the two RNA-motifs with the mode of binding used to bind the original sequence. For linkers ranging from 9 to 4 nt, we did not observe such drastic increase in the violations, meaning that the mode of binding observed with the ISS-N1 (9-nucleotide long spacer) is perfectly compatible with shorter spacers in between the two AGs Altogether, we can conclude that due to topological constraints, the RNA spacer between the two AGs must be at least four nucleotide-long.

**Appendix 2—table 3.** List of the RNAs used for in vitro structural studies and in cell splicing assays.

| ISS-N1 WT | (gga)CCAGCAUUAUGAAAGUGC/A |
| --- | --- |
| ISS-N1-c14g | (gga)CCAGgAUUAUGAAAGUGC |
| ISS-N1-u25g | (gga)CCAGCAUUAUGAAAGgGC |
| ISS-N1-14g25g | (gga)CCAGgAUUAUGAAAGgGC |
| ISS-N1-g13a | CCAaCAUUAUGAAAGUGA |
| ISS-N1-g24a | CCAGCAUUAUGAAAaUGA |
| ISS-N1-13a24a | CCAaCAUUAUGAAAaUGA |
| ISS-N1 −2 | CCAGCAUUA-AAAGUGA |
| ISS-N1 −4 | CCAGCAU-AAAGUGA |
| ISS-N1 −6 | CCAGCA-AAGUGA |
| ISS-N1 −8 | CCAGC-AGUGA |

