## [Decision Letter]

Thank you for submitting your article "Tandem hnRNP A1 RRMs act in concert to repress the splicing of survival motor neuron exon 7" for consideration by *eLife*. Your article has been favorably evaluated by James Manley (Senior Editor) and three reviewers, one of whom, Juan Valcárcel (Reviewer #1), is a member of our Board of Reviewing Editors.

The reviewers have discussed the reviews with one another and the Reviewing Editor has drafted this decision to help you prepare a revised submission.

Joint Report:

Beusch et al. report a detailed structure/function analysis of the interaction between the two RRMs of hnRNP A1 and an intronic silencer present in the SMN genes, involved in splicing regulation. This addresses a long-standing question in protein-RNA recognition and splicing regulation. The authors work out the arrangement of each of the RRM motifs on the bipartite silencer sequence and come to the conclusion that both RRMs contribute additively to RNA binding and that interaction between the two RRM motifs bound to the adjacent half-sites is important for RNA binding and splicing repression.

These results are significant, and may help to identify new binding sites and enhance our understanding of RNA binding and splicing regulation by a paradigmatic RNA binding protein. The results go beyond successive previous structures by crystallography and NMR describing interactions of hnRNP A1 with nucleic acids and establish the architecture of an RNP complex that is the target of an antisense oligonucleotide that has revolutionized the treatment of Spinal Muscular Atrophy. The structural data are of high quality and there is overall good correlation with functional data.

Major issues:

1) The authors characterize HNRNPA1 RRM1 and RRM2 binding to RNA independently by NMR, and identify binding motifs that differ slightly from previous SELEX motifs. The optimal binding sequence for RRM2 was identified as UCAGUU (subsection “Initial binding studies of hnRNP A1 RRMs with small RNA motifs”, Appendix 1—table 1). However, ITC measurements listed in Table 3 show that RRM2 binds to UUAGGUC with 8-fold higher affinity. To investigate how longer strands of RNA interact with UP1 (comprising both RRMs), the authors used a known HNRNPA1 binding site in SMN2 exon 7, with mutations to enhance binding (Figure 3). It is unclear, however, why UP1 R140A, which has reduced RNA binding (Table 3), yielded the strongest signal. The authors explain the ~5-fold increase in RNA-binding affinity of UP1 R1r2, compared to RRM1, by the lack of the inter-RRM linker. Although the mutations in UP1 R1r2 are similar to mutations in UP1 r1R2 that appear not to contribute to RNA binding, the increase in binding could also be explained by residual RNA binding to RRM2 of U1 R1r2, which should be discussed or tested by experiments similar to those listed in Table 3.

2) Further experimental evidence to rule out that multiple HNRNPA1 molecules bind to the same RNA strand could be provided by co-transfecting cells with HNRNPA1 R1r2 and r1R2. If each ISS-N1 binding site can only interact with a single hnRNPA1 molecule, there should not be an additive or synergistic effect.

3) – The protein analysis of Figure 5 are of less quality than the rest of the manuscript. First, the legend of Figure 5 (which should actually be 5A according to the figure display) should state that nuclear lysates (not cell lysates) were used. More important, the authors simply conclude that "The amounts of hnRNP A1 WT and mutants appear comparable", while the amounts of flagged hnRNP A1 are clearly different (e.g. compare lanes 3, 4 and 5) and in general difficult to assess, given the non-linear nature of Western Blot signals and the difficulty to visually normalize by the (also variable) amounts of the DDX5 loading control. It will be important to quantify these differences more accurately (e.g. analyzing by Western Blot serial dilutions of the nuclear extracts) and report also variations between experiments, particularly considering the considerable effort made by the authors to quantify the ratios between splicing isoforms. For example, mutant INT3 appears to be the most active if the three INT mutants, despite the apparently lower levels of protein expression, which may require a structural explanation if the data stand.

4) Based on the result in Figure 5, the authors conclude that RRM2 is more important than RRM1 for ISS-N1 binding and SMN splicing. However, this result is not supported by the data presented in Figure 5, showing that mutations of either 5' AG or 3' AG have similar effects on SMN splicing. In fact, data in Hua et al. 2008 (referenced in the text) show that ASO-mediated steric blocking of the 3' AG (shown here to interact with RRM1) has a stronger effect on SMN2 exon 7 inclusion than steric blocking of the 5'AG (shown here to interact with RRM2). A possible explanation for this discrepancy – related to the previous point – could be different expression levels of hnRNP A1 mutants (based on the loading control) which may account for the difference in SMN1 splicing (Figure 5), rather than RRM2 being more important. If the data are reproducible with comparable R1r2 and r1R2 expression levels, they authors should discuss why mutations of 5' AG and 3' AG in ISS-N1 have the same effect on SMN2 splicing.

5) A manuscript just appeared in PNAS by Tolbert describing the specificity of hnRNPA1. The authors should compare the two sets of results and evaluate them critically.

6) It is a bit unclear if the model of the 2 RRM-RNA structure is a model or a formal structure. This could be clarified, perhaps by showing a table with proper statistics to show what the data actually establish.

*Reviewer #1:*

Beusch et al. report a detailed structure/function analysis of the interaction between the two RRMs of hnRNP A1 and an intronic silencer present in the SMN genes, involved in splicing regulation. The authors work out the arrangement of each of the RRM motifs on the bipartite silencer sequence and come to the conclusion that both RRMs contribute additively to RNA binding and that interaction between the two RRM motifs bound to the adjacent half-sites is important for RNA binding and splicing repression.

These results are important because they differ substantially from previous data describing interactions of hnRNP A1 with nucleic acids and establish the architecture of an RNP complex that is the target of an antisense oligonucleotide that has revolutionized the treatment of Spinal Muscular Atrophy. The structural data are of high quality and there is overall good correlation with functional data.

In my opinion the following revision would improve the manuscript:

The protein analysis of Figure 5 are of less quality than the rest of the manuscript. First, the legend of Figure 5 (which should actually be 5A according to the figure display) should state that nuclear lysates (not cell lysates) were used. More important, the authors simply conclude that "The amounts of hnRNP A1 WT and mutants appear comparable", while the amounts of flagged hnRNP A1 are clearly different (e.g. compare lanes 3, 4 and 5) and in general difficult to assess, given the non-linear nature of Western Blot signals and the difficulty to visually normalize by the (also variable) amounts of the DDX5 loading control. It will be important to quantify these differences more accurately (e.g. analyzing by Western Blot serial dilutions of the nuclear extracts) and report also variations between experiments, particularly considering the considerable effort made by the authors to quantify the ratios between splicing isoforms. For example, mutant INT3 appears to be the most active if the three INT mutants, despite the apparently lower levels of protein expression, which may require a structural explanation if the data stand.

*Reviewer #2:*

Allain and coworkers address a long-standing question in protein-RNA recognition and splicing regulation. The hnRNPA1 protein regulates alternative splicing and microRNA processing and maturation, and has long been studied as a model to understand RNA recognition as well, However, despite successive structures by crystallography and NMR, a structure of the two RRM protein bound to RNA has been missing and, which would come with it, a thorough knowledge of how this protein binds.

Here the authors make several important advances. First, they fully characterize the structure and specificity of each RRM. Then they use this information to optimize the sequence the didomain construct bind to and use this to determine a model of the complete two RRM-RNA structure. Then, they go on to test the biological role of the features discovered in their investigation by using a reporter minigene alternative splicing construct.

The advances are significant on a paradigmatic protein. I have however a few suggestions

1) A manuscript just appeared in PNAS by Tolbert describing the specificity of hnRNPA1. The authors should compare the two sets of results and evaluate them critically.

2) It is a bit unclear if the model of the 2 RRM-RNA structure is a model or a formal structure. This could be clarified, perhaps by showing a table with proper statistics to show what the data actually establish.

3) There is too much detail about NMR throughout the manuscript, which would detract from the attention of many readers. Much could be moved to supplementary material.

*Reviewer #3:*

HNRNPA1 is an abundant RNA-binding protein with important roles in splicing regulation, which involves binding of its two-RRM UP1 domain to RNA splicing silencer elements. Previous results by the Allain lab revealed a different relative orientation of the RRMs in solution, compared to earlier x-ray crystal structures of UP1 bound to telomeric ssDNA. This manuscript characterizes HNRNPA1 RNA binding by solution NMR, and shows that RRM1 and RRM2 can bind to two nearby motifs on the same RNA strand. Cell-based splicing assays with an SMN minigene suggest that disrupting the interaction between RRM1 and RRM2 inhibits HNRNPA1 repressive activity, and both RRMs are necessary for optimal activity. The authors propose a new model for HNRNPA1 binding to RNA, which may help to identify new binding sites.

I do not have the expertise to judge the technical quality of the NMR results, but overall, the data are described clearly and the figures are easy to understand. The results enhance the understanding of RNA binding and splicing regulation by HNRNPA1. Several aspects of the data require clarification:

The authors characterize HNRNPA1 RRM1 and RRM2 binding to RNA independently by NMR, and identify binding motifs that differ slightly from previous SELEX motifs. The optimal binding sequence for RRM2 was identified as UCAGUU (subsection “Initial binding studies of hnRNP A1 RRMs with small RNA motifs”, Appendix 1—table 1). However, ITC measurements listed in Table 3 show that RRM2 binds to UUAGGUC with 8-fold higher affinity. To investigate how longer strands of RNA interact with UP1 (comprising both RRMs), the authors used a known HNRNPA1 binding site in SMN2 exon 7, with mutations to enhance binding (Figure 3). It is unclear, however, why UP1 R140A, which has reduced RNA binding (Table 3), yielded the strongest signal. The authors explain the ~5-fold increase in RNA-binding affinity of UP1 R1r2, compared to RRM1, by the lack of the inter-RRM linker. Although the mutations in UP1 R1r2 are similar to mutations in UP1 r1R2 that appear not to contribute to RNA binding, the increase in binding could also be explained by residual RNA binding to RRM2 of U1 R1r2, which should be discussed or tested by experiments similar to those listed in Table 3.

There appears to be a disconnect between the suggestion that RRM2 is more important for ISS-N1 binding and SMN2 splicing (Figure 5) with the data presented in panels C and D, showing that mutation of either 5' or 3' AG have similar effects on SMN splicing. In fact, data in Hua et al. 2008 (referenced in the text) show that ASO-mediated steric blocking of the 3' AG (shown here to interact with RRM1) has a stronger effect on SMN2 exon 7 inclusion than steric blocking of the 5'AG (shown here to interact with RRM2).

Further experimental evidence to rule out that multiple HNRNPA1 molecules bind to the same RNA strand could be provided by co-transfecting cells with HNRNPA1 R1r2 and r1R2. If each ISS-N1 binding site can only interact with a single hnRNPA1 molecule, there should not be an additive or synergistic effect.

---

## [Author Response]

*Major issues:*

*1) The authors characterize HNRNPA1 RRM1 and RRM2 binding to RNA independently by NMR, and identify binding motifs that differ slightly from previous SELEX motifs. The optimal binding sequence for RRM2 was identified as UCAGUU (subsection “Initial binding studies of hnRNP A1 RRMs with small RNA motifs”, Appendix 1—table 1). However, ITC measurements listed in Table 3 show that RRM2 binds to UUAGGUC with 8-fold higher affinity.*

It is perfectly correct that we did not determine the structure of RRM2 in complex with the RNA yielding the best affinity, but with the RNA giving the best quality of NMR spectra. We tested several RNAs (listed in Appendix 1—table 1), and our choice for the structure determination was based on the NMR exchange regime, as explained in the manuscript in the paragraph “Initial binding studies of hnRNP A1 RRMs with small RNA motifs”. UCAGUU was the sole RNA giving a complex in the so-called slow exchange regime, and was therefore chosen for structure determination. In order to make this point clearer for the reader, we have expanded our initial statement as follows:

“Note that as for RRM1, RRM2 prefers a U at this position (see Table 3), and our complex with RRM2 was thus not determined with the RNA having the best affinity. It had however the optimal exchange properties in terms of NMR and structure determination (see Appendix 1—table 1).”

*To investigate how longer strands of RNA interact with UP1 (comprising both RRMs), the authors used a known HNRNPA1 binding site in SMN2 exon 7, with mutations to enhance binding (Figure 3). It is unclear, however, why UP1 R140A, which has reduced RNA binding (Table 3), yielded the strongest signal.*

As a general principle, we would like to emphasize that exchange regime in NMR complicate the study of complexes. As a result of this, it is not necessarily the complex with the highest affinity that will give the best NMR signals. Similarly, as for the previous point, the choice of the point mutation introduced here (UP1-R140A) was not meant to increase the affinity, but to solely improve the NMR behavior of the complex. In the wild-type complex, some signals at the binding interface disappeared upon RNA addition (Figure 3—figure supplement 2), which indicates a complex in the so-called intermediate exchange regime. Structural investigation with such behavior is not possible since signals at the protein-RNA interface are missing. We thus sought for a complex with improved NMR signals at the interface by minimally mutating either the protein or the RNA or both. It turned out that decreasing the affinity with the R140A mutant lead to improved NMR signals, most probably because it shifted the complex from the unfavorable intermediate exchange regime towards the more favorable fast exchange regime. Most of these NMR specific details have been moved to the Appendices, but in order to make this point clearer for the reader, we have expended our initial statement as follows:

“Since it is not necessarily the complex with the best affinity that will give the best NMR signals, we then decided to take the opposite approach of weakening the overall affinity of the complex with the aim of reaching the fast exchange regime with improved NMR signals.”

In addition, in the main text, this point is now briefly explained as follows:

“Since it is not necessarily the complex with the highest affinity that will give the best NMR signals, we then took the opposite approach of weakening the overall affinity of the complex with the protein mutant UP1-R140A (Table 3).”

*The authors explain the ~5-fold increase in RNA-binding affinity of UP1 R1r2, compared to RRM1, by the lack of the inter-RRM linker. Although the mutations in UP1 R1r2 are similar to mutations in UP1 r1R2 that appear not to contribute to RNA binding, the increase in binding could also be explained by residual RNA binding to RRM2 of U1 R1r2, which should be discussed or tested by experiments similar to those listed in Table 3.*

The reviewers are concerned that the increase in affinity observed for UP1 R1r2 compared to RRM1 could also be explained by residual RNA-binding to RRM2 of UP1 R1r2 despite the mutations in RRM2. In the fitting of ITC data, one of the parameters is the N-value indicative of the stoichiometry of the measured interaction (mentioned on page 17). N is one of the most precisely determined parameters and should be a warning sign for experimental problems, such as a wrong fitting model, when deviations from the expected value do occur (Tellinghuisen, Anal Biochem 2012). Our data robustly fit with a 1:1 stoichiometry (N = 1) (Table 3, [Supplementary-material SD1-data], subsection “RNA specificities and preferences of each individual RRM of hnRNP A1”, last paragraph). Hence, if residual RNA binding to RRM2 occurred we would expect a (substantial) deviation of N from 1. For these reasons we are certain that both in the case of UP1 R1r2 and UP1 r1R2 the introduced mutations strongly dampen RNA binding at the mutated RRM’s binding interface to a level which is too low to perturb the correct measurement of the RNA-binding affinity of the unaffected RRM. Consequently, we are confident that all our data shown in Table 4 represents true dissociation constant measurements for each individual RRM in the context of UP1. In order to make this point clearer for the reader, we have expended our initial statement as follows:

“Importantly, the mutations were efficient in removing the RNA binding ability of the mutated RRM. After fitting with a 1:1 binding model the obtained N-values of our ITC measurements were robustly ~1. This is indicative of a 1:1 stoichiometry of the complex.”

*2) Further experimental evidence to rule out that multiple HNRNPA1 molecules bind to the same RNA strand could be provided by co-transfecting cells with HNRNPA1 R1r2 and r1R2. If each ISS-N1 binding site can only interact with a single hnRNPA1 molecule, there should not be an additive or synergistic effect.*

We thank the reviewers for suggesting this experiment. We have performed it and the results can now be found as Figure 5—figure supplement 4. The co-transfection of both hnRNP A1 R1r2 and hnRNP A1 r1R2 resulted in the same exon 7 inclusion rate as for transfection of hnRNP A1 r1R2 alone. We conclude from this that the two mutants cannot complement each other and that it is therefore unlikely that hnRNP A1 acts as a dimer on the ISS-N1 such as was observed in the crystal structure of hnRNP A1 in complex with telomeric repeat DNA. Together with our other experiments we take this as an additional evidence that what we have observed by NMR is a functional state in the cell.

We comment on these results as follows:

“Our structural data proposes the simultaneous binding of both RRMs to the ISS-N1. […] In support of our model, we could not observe any rescue of hnRNP A1 function when co-transfecting with the two RRM mutants (Figure 5—figure supplement 4).”

*3) The protein analysis of Figure 5 are of less quality than the rest of the manuscript. First, the legend of Figure 5 (which should actually be 5A according to the figure display) should state that nuclear lysates (not cell lysates) were used. More important, the authors simply conclude that "The amounts of hnRNP A1 WT and mutants appear comparable", while the amounts of flagged hnRNP A1 are clearly different (e.g. compare lanes 3, 4 and 5) and in general difficult to assess, given the non-linear nature of Western Blot signals and the difficulty to visually normalize by the (also variable) amounts of the DDX5 loading control. It will be important to quantify these differences more accurately (e.g. analyzing by Western Blot serial dilutions of the nuclear extracts) and report also variations between experiments, particularly considering the considerable effort made by the authors to quantify the ratios between splicing isoforms. For example, mutant INT3 appears to be the most active if the three INT mutants, despite the apparently lower levels of protein expression, which may require a structural explanation if the data stand.*

We acknowledge that the original protein analysis in Figure 5 was of a lesser quality than the rest of the manuscript. The original data was obtained to the best of our knowledge by transfection in HEK293T cells using the calcium phosphate method. During revision, the analysis of the Western Blots revealed non-reproducible overexpression of hnRNP A1 at times between replicates. To manage this issue we changed the protocol for plasmid purification for transfection (now ZymoPURE Plasmid Midiprep Kit, Zymo Research) and as well exchanged the transfection reagent to Lipofectamine 2000. All Western Blots for all replicates have been attached as figure supplements to Figure 5. As can be seen, overexpression of the construct is now reproducible between replicates.

The only mutant that appears to be expressed at a different level under these conditions is hnRNP A1 R1r2. We have tested for dose effects by increasing the protein level by transfection of increased amounts of hnRNP A1 variants (Figure 5—figure supplement 2). We can show that the *SMN* splicing response remains the same under increased protein amounts.

In addition, our Western Blots were recorded by CCD-based detection (Amersham Imager 600, GE Healthcare), which offers a dynamic range of 4.8 orders of magnitude allowing for detection across a wide dynamic range for precise quantitation. This allows for a linear detection over a broad range of signals and should cover our range according to the manufacturer. The data obtained during the revisions are overall very similar to those previously obtained. Differences are precisely described in response to point #4.

Finally, the reviewers have raised the concern that INT3 appears most active, despite the apparently lower levels of protein expression. While with the new transfection protocol INT3 amounts do not differ from INT1 and INT2, the reviewers have raised an interesting question. In Figure 5—figure supplement 1 we show the overall NMR correlation time (τ_c_) of UP1 WT and UP1 INT1-3 mutants. INT3 has, out of the three mutants, a correlation time in the closest range to the wild-type protein. This suggests, that in INT3 the interaction between RRM1 and RRM2 is less disrupted than in INT1 and INT2. This could explain why the INT3 mutant has a behavior closer to the WT protein, explaining its slightly higher activity in the *SMN1* splicing assay. However, as the same cannot be seen for the INT2 mutant, which is also more active in the splicing assay, we refrained from discussing this potential correlation in the paper to avoid over interpretation of our data.

*4) Based on the result in Figure 5, the authors conclude that RRM2 is more important than RRM1 for ISS-N1 binding and SMN splicing. However, this result is not supported by the data presented in Figure 5, showing that mutations of either 5' AG or 3' AG have similar effects on SMN splicing. In fact, data in Hua et al. 2008 (referenced in the text) show that ASO-mediated steric blocking of the 3' AG (shown here to interact with RRM1) has a stronger effect on SMN2 exon 7 inclusion than steric blocking of the 5'AG (shown here to interact with RRM2). A possible explanation for this discrepancy – related to the previous point – could be different expression levels of hnRNP A1 mutants (based on the loading control) which may account for the difference in SMN1 splicing (Figure 5), rather than RRM2 being more important. If the data are reproducible with comparable R1r2 and r1R2 expression levels, they authors should discuss why mutations of 5' AG and 3' AG in ISS-N1 have the same effect on SMN2 splicing.*

Our new sets of experiments with new plasmid preparation and transfection protocols, lead to a different conclusion regarding the relative importance of the single RRMs. Previously, both in the *SMN1* and *SMN2* minigene system, hnRNP A1 r1R2 overexpression consistently had an effect and led to a minor repression of exon 7 inclusion compared to the control transfection with an empty plasmid. With the new protocol in place no such effect can be seen in comparison to the control transfection. Taking the new results into account we had to modify our interpretation of the results and their discussion and have now removed any mention of RRM2 being more important than RRM1. In addition, for the set of experiments where the ISS-N1 spacer length was varied, with the new protocol we no longer observe the increase in exon 7 inclusion for *SMN2* ISS-N1 -2 which we could not explain earlier.

In addition, the reviewers mention the ASO tiling experiment performed by Hua et al. in their 2008 publication where steric blocking of the 3’AG had a stronger effect on SMN2 exon 7 inclusion than blocking of the 5’ AG. We believe that steric blocking will work with a different principle and cannot necessarily be compared to a mutation experiment. In the same publication, a mutational experiment was also performed, where individual positions of the ISS-N1 were mutated. In this case, and like in our experiment, in the absence of hnRNP A1 overexpression, targeting either AG had similar effects on *SMN* splicing.

Finally, the reviewers have raised the concern that a difference in *SMN2* exon 7 splicing could be a result of unequal amounts of overexpressed protein. We have responded to this concern as a part of reviewers’ comment #3.

We understand that each of our experiments only allows us to look at a single part of the UP1:ISS-N1 interaction decoupled from the others. Taking our results together as well as the new experiment added in response to reviewers’ comment #2, we are nevertheless very confident in our interpretation of the experiments.

*5) A manuscript just appeared in PNAS by Tolbert describing the specificity of hnRNPA1. The authors should compare the two sets of results and evaluate them critically.*

Indeed, a manuscript by Tolbert and co-workers appeared in PNAS at the point of submission of our work. Using HTS-EQ and cross-validation with ITC they have found a consensus motif of 5’-YAG-3’ for hnRNP A1. Furthermore, it was found that an additional guanine immediately 3’ to the motif is favored too. Together this matches our findings for a strong preference of 5’-UAG-3’, which can be extended to 5’-U/_C_AGG-3’for a wider description of the preference.

We have added this analysis of the manuscript by Tolbert and co-workers to the Discussion.

*6) It is a bit unclear if the model of the 2 RRM-RNA structure is a model or a formal structure. This could be clarified, perhaps by showing a table with proper statistics to show what the data actually establish.*

The two RRM-RNA structure is not a formal NMR structure, and we refer to it as a “structural model” throughout the manuscript. Although we did use similar protocols as for regular NMR structure calculations, the ensemble that we have obtained cannot be seen as a formal NMR structure. The main reason for this is that the RNA-spacer separating the two RRM-bound motifs of the ISS-N1 was not restricted in our modelling procedure, since we did not assign intermolecular NOEs to this region. For this reason we do not consider it as an actual structure.

In order to make this point clearer for the reader, we have expended the statement in the figure legend of Figure 4 to: “Please note that the path of the RNA-spacer between the 5´ and 3´ motifs is not restrained by experimental constraints. The present structural model therefore illustrates one possible path of the ISS-N1 on the RRMs and should not be seen as the unique conformation adopted by the RNA spacer.”

In addition, in the main text, we have added the following sentence: “Combining all the structural information obtained on this complex, we could build a structural model showing how hnRNP A1 RRMs can assemble on the same pre-mRNA stretch (See Appendix 2 for details).”

And in the Appendix 2 we have added the following paragraph:

“Important remark: It is important to note that this model only illustrates one possible path of the ISS-RNA from RRM2 towards RRM1. […] Nevertheless, this model illustrates that the RNA-spacer is long enough for hnRNP A1 RRMs to loop out RNA and bind simultaneously to the 5´- and 3´-motifs.”